# Spectroscopic glimpses of the transition state of ATP hydrolysis trapped in a bacterial DnaB helicase

Alexander A. Malär [1], Nino Wili [1], Laura A. Völker [1], Maria I. Kozlova [2], Riccardo Cadalbert[1], Alexander Däpp[1], Marco E. Weber [1], Johannes Zehnder [1], Gunnar Jeschke [1], Hellmut Eckert[3,4], Anja Böckmann[5], Daniel Klose [1✉], Armen Y. Mulkidjanian[2,6✉], Beat H. Meier [1✉] & Thomas Wiegand [1,7,8✉]

The ATP hydrolysis transition state of motor proteins is a weakly populated protein state that can be stabilized and investigated by replacing ATP with chemical mimics. We present atomic-level structural and dynamic insights on a state created by ADP aluminum fluoride binding to the bacterial DnaB helicase from *Helicobacter pylori*. We determined the positioning of the metal ion cofactor within the active site using electron paramagnetic resonance, and identified the protein protons coordinating to the phosphate groups of ADP and DNA using proton-detected $^{31}P,^{1}H$ solid-state nuclear magnetic resonance spectroscopy at fast magic-angle spinning > 100 kHz, as well as temperature-dependent proton chemical-shift values to prove their engagements in hydrogen bonds. $^{19}F$ and $^{27}Al$ MAS NMR spectra reveal a highly mobile, fast-rotating aluminum fluoride unit pointing to the capture of a late ATP hydrolysis transition state in which the phosphoryl unit is already detached from the arginine and lysine fingers.

[1] Physical Chemistry, ETH Zürich, Zürich, Switzerland. [2] Department of Physics, Osnabrück University, Osnabrück, Germany. [3] Institut für Physikalische Chemie, WWU Münster, Münster, Germany. [4] Instituto de Física de Sao Carlos, Universidade de Sao Paulo, Sao Carlos, SP, Brazil. [5] Molecular Microbiology and Structural Biochemistry UMR 5086 CNRS/Université de Lyon, Lyon, France. [6] School of Bioengineering and Bioinformatics and Belozersky Institute of Physico-Chemical Biology, Lomonosov Moscow State University, Moscow, Russia. [7] Present address: Max-Planck-Institute for Chemical Energy Conversion, Mülheim an der Ruhr, Germany. [8] Present address: Institute of Technical and Macromolecular Chemistry, RWTH Aachen, Aachen, Germany. ✉email: daniel.klose@phys.chem.ethz.ch; armen.mulkidjanian@uni-osnabrueck.de; beme@ethz.ch; thomas.wiegand@phys.chem.ethz.ch

Adenosine triphosphate (ATP)-driven motor proteins play a key role in various cellular processes[1]. For example, motor proteins belong to the class of ATPases, which hydrolyze ATP into ADP (adenosine diphosphate) and inorganic phosphate to gain chemical energy allowing such enzymes to drive further chemical or mechanical events[2]. Structural insights into the functioning of these molecular machines are not straightforward to obtain, neither by X-ray crystallography, nor by cryo-electron microscopy or NMR spectroscopy due to the difficulty in trapping the intermediate catalytic states occurring during ATP hydrolysis. Diverse ATP analogues can be employed to mimic different stages of ATP hydrolysis as closely as possible[3,4] which, in combination with molecular dynamics simulations[5], can give mechanistic insights into complex biomolecular reaction coordinates. Of particular interest in unravelling the ATP hydrolysis reaction mechanism is the transition state of the phosphoryl ($PO_3^-$) transfer reaction (see Fig. 1 for a sketch of the limiting case of an associative ATP hydrolysis mechanism[6]). Metal fluorides have been found to mimic such states for structural studies, mostly using X-ray crystallography[7,8]. The number of deposited protein structures containing analogues such as $AlF_4^-$, $AlF_3$ and $MgF_3^-$ has strongly increased in the past years[7]. $AlF_4^-$ forms together with the phosphate oxygen atom of ADP as well as an apical water molecule an octahedral complex mimicking the "in-line" anionic transition state of phosphoryl transfer, whereas $AlF_3$ and $MgF_3^-$ form trigonal-bipyramidal complexes[7]. The formation of $AlF_4^-$ or $AlF_3$ is controlled by pH, the latter being favored at lower pH values[9]. However, some concern regarding their discrimination, e.g. distinction between $AlF_3$ and $MgF_3^-$, has been raised and it could be shown by $^{19}F$ NMR spectroscopy that some complexes, which were believed to contain $AlF_3$, contain instead $MgF_3^-$ [9]. Similarly, in lower resolution X-ray structures (>2.8 Å) $AlF_3$ cannot be distinguished unambiguously from $AlF_4^-$ [9,10].

We present magnetic resonance approaches using EPR and solid-state NMR to obtain spectroscopic insights into the transition state of ATP hydrolysis which we trap for the oligomeric bacterial DnaB helicase from *Helicobacter pylori* (*Hp*, monomeric molecular weight 59 kDa) by using the transition-state analogue $ADP:AlF_4^-$. The motor domain of the helicase belongs to P-loop fold nucleoside-triphosphatases (P-loop NTPases), one of the largest protein families, which includes motor proteins like myosins, kinesins, and rotary ATPases. About 10–20% of genes in any genome encode for diverse P-loop NTPases[11]. In these enzymes, ATP or guanosine triphosphate (GTP) molecules are bound to the so-called Walker A motif GxxxxGK[S/T] of the signature P-loop of the motor nucleotide-binding domain (NBD)[12,13]. Bacterial DnaB helicases, which use the energy of ATP hydrolysis to unwind the DNA double helix, belong to the ASCE division of P-loop NTPases. The members of this division are characterized by an additional β-strand in the P-loop and a catalytic glutamate (E) residue next to the attacking water molecule[14–16]. Within the ASCE division, DnaB helicases are attributed to the RecA/F1 class[16]. Generally, P-loop NTPases need to be activated before each turnover because otherwise, they

would promptly consume the entire cellular stock of ATP and GTP. As inferred from the comparative structure analysis of NTPases with transition-state analogues, such as $NDP:AlF_4^-$ or $NDP:MgF_3^-$, the activation is mostly achieved by the insertion of a positively charged hydrolysis-stimulating moiety (usually, a positively charged arginine or lysine "finger" or a potassium ion) between the α- and γ-phosphates[7,17–19]. As shown by MD simulations, linking of α- and γ-phosphates by the stimulating moiety leads to rotation of the γ-phosphate group yielding a hydrolysis-prone conformation of the triphosphate chain[20]. In DnaB helicases, activation is achieved through the interaction with the neighboring domain that provides a pair of hydrolysis-stimulating Arg and Lys residues. RecA-type ATPases generally differ from most other P-loop NTPases in that their stimulating residues, which operate in a tandem, interact upon activation only with the γ-phosphate, but not with the α-phosphate group. This interaction triggers the ATP hydrolysis; the triggering mechanism, however, has yet to be determined[21].

*Hp*DnaB belongs to the class of SF4-type, ring-shaped DnaB helicases, and only two crystal structures in complex with ATP analogues have been reported so far, namely the one from *Aquifex aeolicus*[22] (*Aa*DnaB, PDB accession code 4NMN) *Bacillus stearothermophilus*[21] (currently *Geobacillus stearothermophilus*, *Bst*DnaB, PDB 4ESV). *Aa*DnaB is complexed with ADP only and *Bst*DnaB with $GDP:AlF_4^-$ as well as single-stranded DNA, which is similar to our $ADP:AlF_4^-$ complex that we study in presence and absence of DNA herein.

W-band electron–electron double resonance (ELDOR)-detected NMR (EDNMR)[23–26] and electron–nuclear double resonance (ENDOR)[27,28] allow for the positioning of the divalent metal ion within the active site by identifying nuclei in its vicinity. The native $Mg^{2+}$ co-factor is replaced for such studies by the EPR-observable paramagnetic $Mn^{2+}$ analogue[29] (the biological functionality is maintained to about 80% under such conditions compared to the one observed in presence of $Mg^{2+}$)[30]. $^{19}F$, $^{27}Al$ and $^{31}P$ nuclear resonances were observed among others in the EDNMR spectra, proving the binding-mode of $ADP:AlF_4^-$. The extracted $^{31}P$ hyperfine coupling constants and the detection of $^{19}F$ and $^{27}Al$ nuclei in the proximity of the co-factor point to a coordination of the $Mn^{2+}$ ion to the β-phosphate of ADP and $AlF_4^-$.

Solid-state NMR can identify the amino-acid residues involved in the coordination of the ATP analogue. Protons are of particular interest as their resonance frequencies can contain information regarding their engagement in hydrogen bonds. Fast magic-angle spinning (MAS) nowadays provides sufficient spectral resolution for proton-detected sidechain studies[31]. Indeed, proton detection at fast MAS has become an important tool in structural biology in the past years for unraveling protein structures[32–41], to characterize RNA molecules[42] and protein–nucleic acid interactions[43–45], and to address protein dynamics[46–52]. A key advantage of solid-state NMR is the straightforward sample preparation, which simply consists of sedimentation from solution into the solid-state NMR rotor without requiring crystallization steps[53,54] yielding long-term

**Fig. 1 Sketch of the associative ATP hydrolysis mechanism with a trigonal-bipyramidal transition state.** ‡ indicates the transition state.

stable protein samples[55]. A further advantage is the sensitivity of NMR in identifying hydrogen bonds, which is often not achievable by standard structure determination techniques, such as X-ray crystallography or cryo-electron microscopy (EM), in which resolution in the order of 1 Å (for cryo-EM a slightly lower resolution might be sufficient[56]) has to be achieved. We have already previously reported that the transition state of ATP hydrolysis is accessible for DnaB by employing the ATP-analogue ADP:AlF$_4^-$ [57], but the direct identification of hydrogen bonds required for characterizing the noncovalent interactions driving molecular recognition of both, ATP and DNA, was hardly possible and only spatial proximities derived from $^{31}$P–$^{13}$C/$^{15}$N correlation experiments or the proton chemical-shift values were explored[44].

We herein identify protein residues engaged in hydrogen bonding to the phosphate groups of nucleotides (ADP:AlF$_4^-$ and DNA) by (i) measuring high-frequency shifted proton resonances characteristic for hydrogen-bond formation[58], (ii) probing spatial proximities in dipolar-coupling based proton-detected $^{31}$P,$^1$H correlation experiments at fast MAS (105 kHz) and (iii) using the temperature dependence of $^1$H chemical-shift values as a probe for hydrogen bonding, an approach well known in solution-state NMR[59–61], and recently extended to the solid state[62]. Note, that we herein report a proton-detected $^{31}$P,$^1$H correlation spectrum at fast MAS frequencies using a sub-milligram sample amount, which was so far, to the best of our knowledge, not possible with any of the previous equipment. This is an important step for proving hydrogen bonding in protein–nucleic acid complexes ranging from proteins involved in DNA replication or virus assemblies by solid-state NMR and to derive nucleotide-binding modes, even in quite large systems as the one we looked at. From a combination of (i)–(iii), key contacts between the ADP phosphate groups and residues located in the Walker A motif were identified, as well as two hydrogen bonds to the phosphate groups of the two DNA nucleotides. To complement our spectroscopic characterization of the ATP hydrolysis transition state, we performed $^{19}$F and $^{27}$Al MAS experiments to access information about bound AlF$_4^-$. The spectra indicate a fast rotation of the AlF$_4^-$ unit implying that AlF$_4^-$ is not rigidified by coordinating protein residues indicating that the ADP:AlF$_4^-$ trapped state of DnaB possibly describes a late transition state, just after the bond fission, but before the release of the phosphate group from the catalytic pocket.

## Results

### EPR enables the positioning of the metal ion co-factor within the active site.
Binding of ADP:AlF$_4^-$ to the protein is revealed in EDNMR experiments, which employ the hyperfine couplings between a paramagnetic center and nearby nuclei to detect the latter. EDNMR has been used to characterize transition states of ATP hydrolysis, often in the context of ABC transporters for which such a state is successfully mimicked by ADP-vanadate[63,64]. Figure 2a shows the Mn$^{2+}$ EDNMR spectrum of DnaB complexed with ADP:AlF$_4^-$ (red) compared to the reference spectrum of DnaB complexed only with ADP (cyan), here using a non-$^{13}$C/$^{15}$N labeled protein (see below for $^{13}$C/$^{15}$N labeling). While in both spectra couplings to $^{31}$P nuclei are observed, additional peaks for $^{19}$F and $^{27}$Al are detected only for the ADP:AlF$_4^-$ bound state consistent with the presence of AlF$_4^-$ in the NBD of DnaB. The weakly coupled $^{19}$F is also observed by ENDOR (see Supplementary Fig. 2 for the spectrum and discussion of extracted parameters) and indicates its proximity to Mn$^{2+}$. To corroborate that these resonances are due to DnaB-bound Mn$^{2+}$:ADP:AlF$_4^-$ and to rule out that these correlations originate from the formation of the Mn$^{2+}$:ADP:AlF$_4^-$ complex in

solution, we recorded EDNMR spectra on a frozen control solution in the absence of protein and indeed we do not observe any $^{19}$F and $^{27}$Al resonances (purple spectrum in Fig. 2a). Interestingly, in the presence of protein, two groups of $^{31}$P resonances are detected: a hyperfine-split doublet (denoted $^{31}$P$^d$ in Fig. 2a) and an unresolved doublet (denoted $^{31}$P$^u$). Davies $^{31}$P Electron-Nuclear DOuble Resonance (Davies ENDOR)[27] experiments were performed on the Mn$^{2+}$-containing protein complex (Fig. 2b) to extract the hyperfine tensor $A$ of the doublet. Line shape simulations yield a large $A_{iso}$ value of 4.7 MHz (for all $^{31}$P hyperfine tensor parameters extracted from the spectrum see Supplementary Table 1). This value is similar to published values for an ADP:Mn$^{2+}$ complex in which the Mn$^{2+}$ ion binds symmetrically to the two ADP phosphate groups[65,66] or an ATP:Mn$^{2+}$ complex[67]. Mims ENDOR[28] experiments were performed to detect the small $A_{iso}$ value of the in EDNMR unresolved doublet which is determined to be 0.3 MHz (see Fig. 2c). Mims ENDOR measurements on the control solution did not show this doublet. We assign the large $A_{iso}$ value (4.7 MHz) to $^{55}$Mn–$^{31}$Pβ and the small $A_{iso}$ value (0.3 MHz) to $^{55}$Mn–$^{31}$Pα hyperfine couplings indicating that the Mn$^{2+}$ ion is located much closer in space to the Pβ atom of ADP than to the Pα atom. This assignment is supported by Density Functional Theory (DFT) calculations of the hyperfine coupling tensors performed on small clusters mimicking the Mn$^{2+}$ coordination sphere extracted from the available crystal structures of SF4 helicases (*Bst*DnaB:GDP:AlF$_4^-$:DNA[21] and *Aa*DnaB:ADP[22]) although it has to be noted that the uncertainty in the exact metal ion position due to insufficient resolution of the electron density and the initial presence of Ca$^{2+}$ instead of Mg$^{2+}$ in the 4ESV structure might be significant and influence the results of the calculations (Supplementary Table 1 and Supplementary Fig. 2).

We additionally performed EDNMR experiments using uniformly $^{13}$C/$^{15}$N labeled DnaB complexed with Mn$^{2+}$:ADP:AlF$_4^-$. While the same $^{19}$F, $^{27}$Al and $^{31}$P features discussed above are present in the spectrum, additional intense $^{13}$C and $^{15}$N resonances are observed (Supplementary Fig. 3). In combination with the absence of such resonances in the corresponding reference spectrum measured in the absence of protein (Supplementary Fig. 3), this provides further evidence for binding of Mn$^{2+}$:ADP:AlF$_4^-$ to DnaB. Note that the EPR experiments were performed on the protein complex in absence of DNA in contrast to most solid-state NMR experiments described below. As described in earlier work, the ADP:AlF$_4^-$ states in presence and absence of DNA are highly similar[57] and we thus recorded EPR experiments only on one of these complexes.

### Hydrogen bonds to the phosphate groups of ADP and DNA nucleotides identified by fast MAS experiments.
Solid-state NMR experiments on DnaB complexed with ADP:AlF$_4^-$ and single-stranded DNA (a polythymidine stretch with 20 DNA nucleotides was used[68]) allow a direct view into the NBD. Figure 3a shows the previously reported $^{31}$P-detected cross-polarization (CP)-MAS spectrum of DnaB in complex with ADP:AlF$_4^-$ and DNA (see Fig. 3b for the atomic numbering) recorded at 17 kHz MAS[57]. Two narrow resonances are detected for both, the Pα and Pβ of ADP (at −6.0 and −7.1 ppm, respectively) as well as for the DNA phosphate groups (at 0.5 and −1.1 ppm). The latter observation reflects that two DNA nucleotides bind to one DnaB monomer, which is characteristic for SF4-type helicases[44,57]. Proton-detected NMR experiments at fast MAS frequencies (>100 kHz) allow the identification of protons engaged in hydrogen bonds requiring only small amounts of protein in the order of 0.5 mg. The $^1$H NMR chemical-shift value serves as a sensitive indicator for the

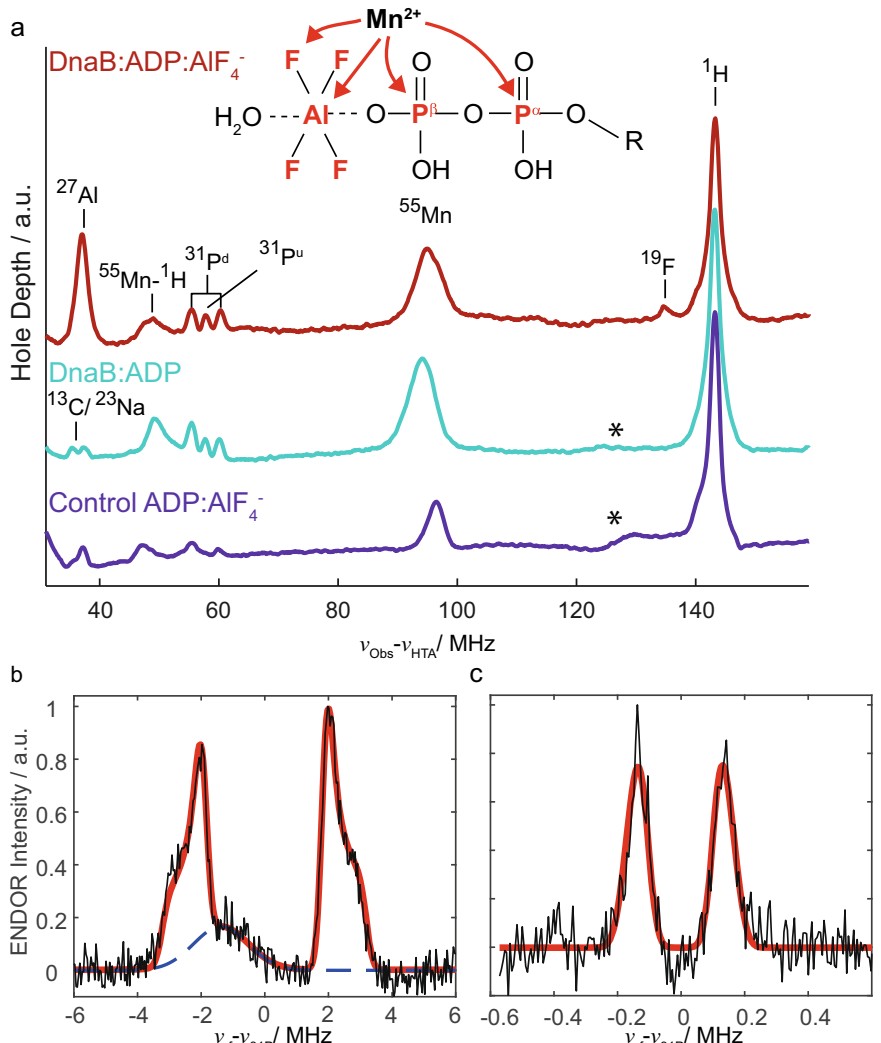

**Fig. 2 EPR characterizes binding of the metal ion co-factor and ADP:AlF$_4^-$ to DnaB. a** EDNMR highlighting hyperfine couplings and thus proximities between the Mn$^{2+}$ metal center and surrounding nuclei measured for DnaB:ADP:AlF$_4^-$ (red) and DnaB:ADP (cyan), as well as for a control solution containing only Mn$^{2+}$:ADP:AlF$_4^-$ in the same buffer used for the protein sample (purple). The assignments of the peaks to the nuclear resonance frequencies are shown. The cyan spectrum is reproduced from ref. [64]. Asterisks mark a broad, currently unassigned feature that is possibly due to Mn double quantum or combination lines[23]. $^{31}$P Davies ENDOR (**b**) and Mims ENDOR (**c**) recorded on DnaB:ADP:AlF$_4^-$. The red lines represent line shape simulations using EasySpin[106] based on $A_{iso}$-values of 0.3 and 4.7 MHz (for all parameters see Supplementary Table 1). The broad background peak in (**b**) is most likely a third harmonic of one of the Mn$^{2+}$ hyperfine lines and was removed for fitting.

formation of hydrogen bonds: a de-shielding effect is observed if protons are engaged in such interactions[44,45,58,69]. However, the chemical shift alone is not a sufficient criterion to prove hydrogen bonding. We therefore extend the experimental approaches to directly detecting such interactions by the presence of through-space $^{31}$P,$^1$H dipolar couplings in hPH correlation experiments at 105 kHz MAS. The hPH spectra were recorded with two different $^1$H–$^{31}$P CP contact times (1.5 and 3.5 ms) on a $^{13}$C,$^{15}$N uniformly labeled, deuterated and 100% back-exchanged sample of DnaB in which the ADP and the DNA remained at natural abundance. Note that this deuterated version of the protein has been chosen over a fully protonated sample to increase the intrinsically rather low signal-to-noise ratio in such a large protein due to the narrowing of the proton resonances by roughly a factor of three attributed to the dilution of the proton dipolar network (see Supplementary Fig. 4 for the proton line-widths determined for a deuterated and fully protonated sample)[70]. Figure 3c shows the rather sparse 2D hPH correlation spectrum (with 3 ms CP contact time) of the DnaB complex and indeed protein–phosphate

correlations to all four $^{31}$P resonances observed in Fig. 3a are visible. The CP-based hPH experiment proves spatial proximities between proton nuclei in the vicinity of the phosphate groups. An INEPT-based experiment transferring polarization directly over the hydrogen bond via the $J$-couplings (typical $^2J(^{31}$P–$^1$H) values are in the order of 3 Hz[71,72]) was not successful due to a too short proton transverse relaxation time compared to the required INEPT transfer delay period (see Supplementary Fig. 5). The resonance assignments shown in Fig. 3c and Supplementary Fig. 6 (CP contact time of 1.5 ms) were obtained using the deposited proton chemical-shift values (BMRB accession code 27879). The hPH spectra reveal intense signals and thus spatial correlations between Pβ of ADP and S206, G208, K209 and T210, all located in the conserved Walker A motif of the P-loop in the motor domain of the helicase[73]. Note that for all mentioned amino acids correlations to the backbone amide protons are observed, except for K209 for which additional sidechain Hζ protons are detected. For the Pα resonance of ADP only weak correlations are observed, the strongest one to S211 and an unassigned resonance,

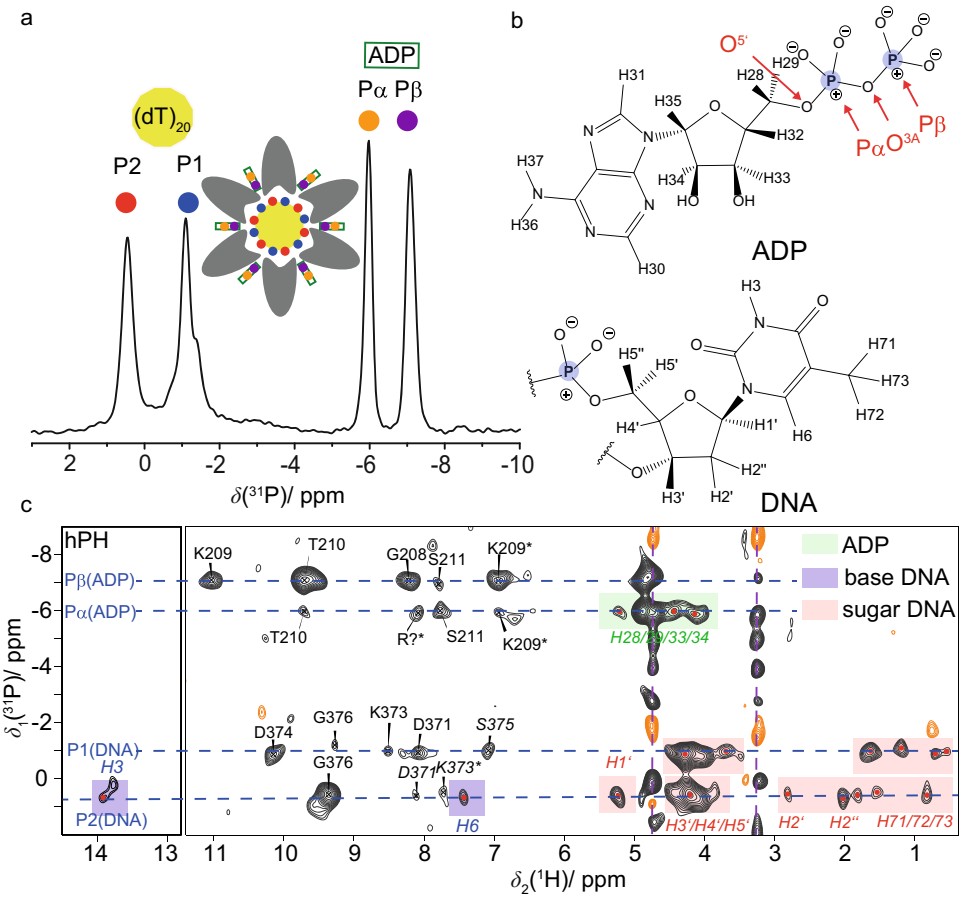

**Fig. 3 ADP and DNA recognition in DnaB highlighted by phosphorus-proton contacts identified at fast MAS. a** $^1$H → $^{31}$P (hP) CP-MAS spectrum of DnaB:ADP:AlF$_4^-$:DNA adapted from ref. [57] (http://creativecommons.org/licenses/by/4.0/) showing the resonance assignments of the DNA and ADP phosphate groups. The shoulder in the $^{31}$P resonance at ∼−1.4 ppm possibly results from rigidified DNA nucleotides, which are, however, not coordinating to DnaB. **b** Chemical structures of ADP and DNA (thymidine) molecules including the numbering of proton atoms following the convention of the BMRB database (DNA) and the recent IUPAC recommendations for nucleoside phosphates[107]. Phosphorus atoms are highlighted in blue. **c** CP-based hPH correlation spectrum (CP contact time 3 ms) recorded on DnaB in complex with ADP:AlF$_4^-$ and DNA at 20.0 T external magnetic field and 105 kHz MAS. The protein resonance assignment is taken from ref. [44] (BMRB accession code 27879). Regular-printed residue labels: Chemical-shift deviation to reported proton shifts <0.05 ppm. Italic-printed residue labels: Chemical-shift deviation to assigned proton shifts ≥0.05 ppm. All proton shifts are assigned to amide backbones, except the ones indicated by an asterisk, which are associated to sidechain atoms. Correlations between the phosphate groups and ADP or DNA are highlighted in green and light red/purple, respectively. The assignments of the DNA proton resonances are based on average chemical-shift values reported in the BMRB database (www.bmrb.wisc.edu). The pink dashed lines highlight signals from insufficiently suppressed DNA in solution.

possibly an arginine residue, which has been detected in previous NHHP experiments[44] (R242 or the "arginine finger" R446 from a neighboring DnaB subunit). The main difference in the spectrum recorded at shorter CP contact times (Supplementary Fig. 7) is that correlations to the ADP and DNA protons (sugar and base) present in the spectrum recorded at 3.5 ms contact time (high-lighted in light red and green in Fig. 3c) are absent. It is important to note that the herein described hPH experiments appear to be much more selective for detecting direct coordination partners than the previously described $^1$H–$^1$H spin-diffusion based NHHP and CHHP experiments[74] and possibly also TEDOR experiments[75], thereby providing a more detailed picture of the local geometry around the phosphate groups of ADP and DNA[4,76] than reported previously[44]. The hPH spectrum in Fig. 3c also contains important information regarding the DNA coordination. Actually, only two intense backbone amide correlations to the two DNA phosphate groups, D374 in case of P1 and G376 in case of P2 are observed. Together with our previous observation of K373 forming a salt-bridge to P2 via the lysine sidechain, only three contacts seem to coordinate the DNA in this molecular recognition process.

The high-frequency shifts of their amide protons and their spatial proximity to the phosphate ADP group already point to the engagement of K209 and T210 in hydrogen bonding as discussed above. To further verify this, we determined the temperature dependence of their chemical shifts between 294 and 302 K (sample temperatures, see Methods Section). Due to their characteristic chemical shifts (and thus their isolated position in the 2D fingerprint spectrum) the temperature dependences could be directly extracted from 2D CP hNH experiments. It is well known from solution-state NMR that the chemical shifts of protons in strong intramolecular hydrogen bonds experience only a weak temperature dependence[60,61] as recently also shown by solid-state NMR[62]. However, for protons in rather weak hydrogen bonds, the resonances become significantly more shielded upon increasing the temperature, due to an increase in the average hydrogen-bond length. Figure 4 shows the tempera-ture dependence for residues identified in the hPH spectra (left column). Indeed, K209 and T210, previously identified as forming hydrogen bonds to the Pβ of ADP, show an almost vanishing temperature coefficient (slope of the corresponding linear regression). Similar values are found for D374 and G376

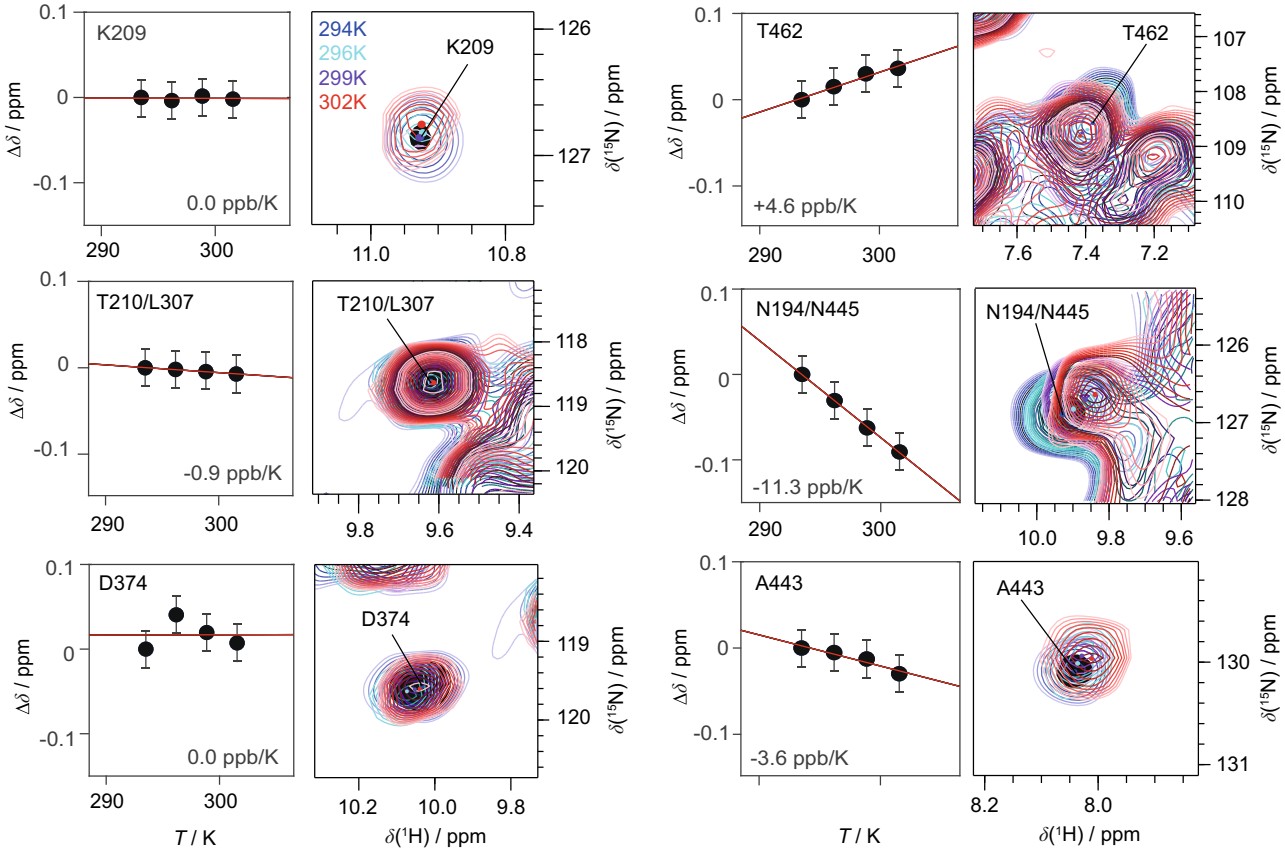

**Fig. 4 Temperature-dependent proton chemical-shift values as indicators for hydrogen-bond formation.** Residue-specific temperature coefficients and corresponding temperature-dependent hNH spectra (based on two $^1$H,$^{15}$N CP steps) recorded at 20.0 T with a spinning frequency of 100 kHz for deuterated and 100% back-exchanged DnaB complexed with ADP:AlF$_4^-$ and DNA. Temperature-dependent chemical-shift deviations (black circles) are referenced to the corresponding value at 294 K sample temperature. Chemical-shift values were extracted from $n = 1$ experiments and are represented as $\delta \pm 0.05$ ppm, where the shown error bar represents an estimate of the expected uncertainty within such experiments.

(Supplementary Fig. 7) assumed to be involved in DNA coordination. In contrast, Fig. 4 (right column) shows resonances associated with a larger temperature coefficient thus not being involved in hydrogen bonds (for all extracted temperature coefficients see Supplementary Fig. 7).

**Solid-state NMR shows that the AlF$_4^-$ unit is highly mobile.** The AlF$_4^-$ unit can be detected in $^{19}$F- and $^{27}$Al-detected MAS experiments. Figure 5a displays the $^{19}$F MAS spectrum of DnaB:ADP:AlF$_4^-$ in the presence and absence of DNA. Interestingly, only one $^{19}$F resonance line at around −146 ppm is detected for the protein-bound AlF$_4^-$ group pointing to a fast chemical-exchange process, most probably a rotation of the unit (*vide infra*, for the $^{19}$F spectrum in the absence of protein see Supplementary Fig. 8). The additional sharp $^{19}$F resonances visible in the spectra are attributed to the excess of AlF$_4^-$ and related species present in the supernatant of the NMR rotor (roughly 50 weight percent after sedimentation[77]). Around 4% of the AlF$_4^-$ remains in the supernatant after the rotor-filling step. The resonance assignments displayed in Fig. 5a are based on reported solution-state NMR assignments[78,79]. A similar chemical-exchange process has also been observed for the RhoA/GAP:GDP:AlF$_4^-$ complex[80], for the GTPase hGBP1[81] and for the motor protein myosin[78] in solution-state $^{19}$F NMR experiments.

The $^{27}$Al satellite transition NMR spectrum (SATRAS, Fig. 5b) of the sideband family is observed at $\delta_{iso} = -0.2$ ppm pointing to an octahedral coordination geometry of the $^{27}$Al nucleus[82]. The spectrum allows to extract the quadrupole coupling

constant (C$_Q$) which amounts to only ~570 kHz. The central m = 1/2↔m = −1/2 transition is observed at a similar resonance shift indicating a small contribution of the second-order quadrupolar shift. The C$_Q$-value is significantly lower than expected for a six-fold oxygen/fluorine coordinated aluminum species. C$_Q$-values for crystalline aluminum hydroxyfluorides are typically in the order of 5 MHz[83]. We attribute this effect to a rotation on the NMR time scale of the AlF$_4^-$ unit around an axis inclined by an angle θ with respect to the direction of the principal component of the electric field gradient tensor (V$_{zz}$, see Fig. 5c). The angle θ must be close, about 5–10°, to the magic angle (54.7°) leading to the significant reduction of the anisotropy of the quadrupolar interaction (see Fig. 5d). Similar observations were made for the DnaB complex in the absence of DNA (see Supplementary Fig. 9). Alternatively, the reduction of the quadrupolar interaction could be achieved by a rotational diffusion process, in which the angle θ varies randomly and is on average close to the magic angle. Note that the coordination of AlF$_4^-$ to the β-phosphate of ADP is also reflected in a low-frequency shift of the corresponding $^{31}$P ADP resonance (−4.5 ppm compared to the ADP-bound state[57]), which is a similar trend as observed for aluminophosphate gels and glasses[84].

The rotational motion or even diffusion of this unit (with a correlation time shorter than the inverse quadrupolar coupling constant) reflects the absence of tight binding either to the protein (e.g. via hydrogen bonds to the fluorine atoms) or to the metal ion co-factor. The $^{27}$Al isotropic chemical-shift value of close to 0 ppm is characteristic for an octahedrally coordinated Al-species,

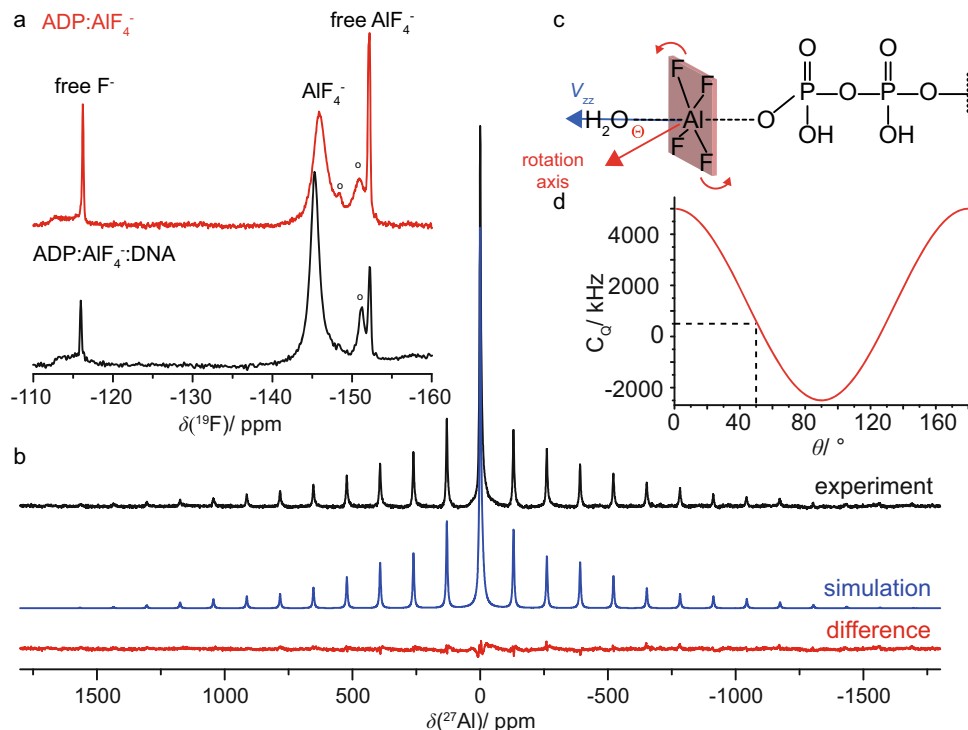

**Fig. 5 The AlF$_4^-$ species bound to DnaB is rotating. a** [19]F MAS-NMR spectra recorded at 14.0 T with a MAS frequency of 17.0 kHz and with the EASY background suppression scheme[108]. Spectra were acquired on DnaB:ADP:AlF$_4^-$ in the presence and absence of DNA; "o" indicates precipitated AlF$_x$(OH)$_{6-x}$ species. **b** [27]Al MAS-NMR spectrum of DnaB:ADP:AlF$_4^-$:DNA recorded at 11.74 T with a spinning frequency of 17.0 kHz (black) and corresponding line shape simulation using DMFIT (version 2019)[109] assuming C$_Q$([27]Al) = 570 kHz, η$_Q$([27]Al) = 0.98, Δ$_σ$([27]Al) = −186 ppm and η$_σ$([27]Al) = 0.64. The central resonance is fitted with two additional Lorentzian lines possibly originating from aluminum hydroxyl fluorides in solution[79]. The difference spectrum is shown in red. **c** Schematic illustration of the rotation of the AlF$_4^-$ molecule. θ describes the angle between the rotation axis and the principal component ($V_{zz}$) of the [27]Al electric field gradient pointing along the O···Al···O axis. **d** Calculation of the effective [27]Al quadrupolar coupling constant according to C$_Q$ = C$_Q$(static) · $\frac{1}{2}$ (3cos$^2$θ − 1), assuming a static C$_Q$ value of 5 MHz (AlF$_4$O$_2$ species in AlF$_x$(OH)$_{3-x}$ ·H$_2$O as taken from reference [83]). Fast rotation of the AlF$_4^-$ unit on the NMR time scale is assumed in this calculation.

in our case most likely formed by four fluoride ligands, one oxygen ligand from the ADP phosphate backbone and one water molecule originating from an "in-line" geometry of phosphoryl transfer[7] or the catalytic glutamate as observed in the *Bst*DnaB structure[21].

**Homology modelling points to a free rotating AlF$_4^-$ detached from lysine and arginine fingers in SF4 helicases.** We performed homology modelling based on the available bacterial helicase structures to investigate whether the dynamic behaviour of the AlF$_4^-$ moiety could be related to the activation mechanism of RecA NTPases. Although the crystal structure of the *Hp*DnaB dodecamer is available (PDB accession code 4ZC0[85]), its low resolution of 6.7 Å and the absence of either DNA or of bound nucleotides prevents its use for modelling the ADP:AlF$_4^-$ interactions in the catalytic site of a DNA-bound protein. Therefore, we reconstructed the mechanism of the activation from analysis of the DNA- and Ca$^{2+}$:GDP:AlF$_4^-$-containing *Bst*DnaB structure (PDB accession code 4ESV, resolution 3.2 Å)[21]. In the *Bst*DnaB structure, the Ca$^{2+}$:GDP:AlF$_4^-$ moieties are bound to five out of six catalytic centres (Supplementary Fig. 10). Furthermore, the positions and orientations of the AlF$_4^-$ moieties differ among the five catalytic sites (Supplementary Fig. 10). By considering these different configurations as mimics of different reaction intermediates, the reaction steps could be reconstructed in the following way: Generally, the interaction with a stimulating moiety enables the nucleophilic attack on the γ-phosphate group by an apically positioned water molecule[17]. In numerous P-loop NTPases this step manifests itself in formation of pre-

transition-state analogue complexes NDP:AlF$_4^-$:H$_2$O$_{cat}$ or NDP:MgF$_3^-$:H$_2$O$_{cat}$ (for recent reviews see refs. [7,8]). However, such a state with an apically placed H$_2$O$_{cat}$ is not observed in any of the five AlF$_4^-$-containing sites of the *Bst*DnaB structure. Therefore, in Fig. 6a, we model this transition state using two structures as templates, the ADP:AlF$_4^-$:H$_2$O$_{cat}$ structure from the ABC-NTPase of the *E. coli* maltose transporter (which belongs to the same ASCE division as DnaB), as well as the whole structure of the closely related ADP:AlF$_4^-$-containing RecA of *E. coli* (with the anticipated catalytic water molecule unresolved). As seen in Fig. 6a, the stimulating Arg and Lys residues in RecA form H-bonds with two fluorine atoms of AlF$_4^-$ (blue dashed lines). Comparison of the AlF$_4^-$ positions in different monomers of the *Bst*DnaB structure, as shown in Fig. 6b, c and Supplementary Fig. 8, suggests that Arg and Lys residues are able, together, to twist/tilt the γ-phosphate group, which is mimicked by AlF$_4^-$ in Fig. 6b, c. While in Fig. 6a, b the stimulating Lys and Arg residues are H-bonded to AlF$_4^-$, its further movement away from the nucleotide, as seen in Fig. 6c, leads to the weakening of H-bonds or even their entire dissociation (note the longer distances indicated in Fig. 6c), possibly yielding an almost unbound AlF$_4^-$ unit tilted relative to its catalytic position (compare Fig. 6c with Fig. 6a) in agreement with our solid-state NMR observations of a nearly freely rotating AlF$_4^-$ moiety.

## Discussion

The transition state of ATP hydrolysis in the bacterial DnaB helicase from *Helicobacter pylori* has been trapped by using the mimic ADP:AlF$_4^-$. Such metal fluorides have been successfully

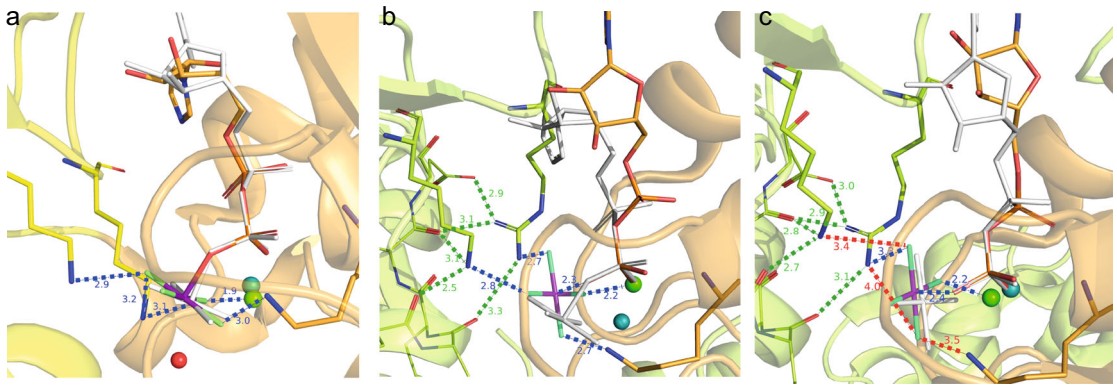

**Fig. 6 The transition-state analogue AlF$_4^-$ can adopt different orientations in diverse P-loop ATPases of the ASCE division. a** AlF$_4^-$ binding in RecA from *E. coli* (PDB accession code 3CMW[110]). The P-loop domain (NBD domain) is shown in orange, the NBD domain of the adjacent activating subunit that provides the stimulating "fingers" is colored yellow. The magnesium ion is shown as a green sphere. To show the catalytic water molecule H$_2$O$_{cat}$, the ADP:AlF$_4^-$ complex is superimposed with the structure of the ADP:AlF$_4^-$:H$_2$O$_{cat}$ complex from the ABC ATPase of the maltose transporter MalK (see PDB accession code 3PUW[111]). The ADP molecule and AlF$_4^-$ of MalK are shown in white, H$_2$O$_{cat}$ as a red sphere, Mg$^{2+}$ as a teal sphere. ADP moieties were superimposed using atoms O$^{3A}$, P$^B$ and O$^{3B}$ (see Fig. 3b for the atom notation used for ADP) in Pymol[112]. **b** Coordination of AlF$_4^-$ in the *Bst*DnaB structure (see Supplementary Fig. 10, PDB ID 4ESV and ref. [21]). The orange, nucleotide-binding chain and the green activating chain correspond to the chains C and B of the 4ESV[21]. To show the displacement of AlF$_4^-$, the GDP:AlF$_4^-$ complex is superimposed, as described for panel **a**, with ADP:AlF$_4^-$ bound to the RecA protein (PDB ID 3CMW, see panel **a**). The H-bonds formed by AlF$_4^-$ are shown in blue, the additional interactions that stabilize the position of stimulating sidechains of K418 and R420 are shown in green. The AlF$_4^-$ moiety is twisted in comparison to the transition-state-mimicking complex shown on panel **a**. **c** Coordination of AlF$_4^-$ in the structure of *Bst*DnaB (see Supplementary Fig. 10, PDB ID 4ESV[21]). The orange, nucleotide-binding chain and the green activating chain correspond to the chains F and E of the 4ESV PDB structure[21]. To show the further displacement of AlF$_4^-$, the GDP:AlF$_4^-$ complex of subunit F is superimposed, as described for panel **a**, with the same complex bound to subunit C of the 4ESV PDB structure[21], which is white coloured (see panel **b**). Bonding interactions that are observed for the GDP:AlF$_4^-$ complex trapped at the B/C interface (see panel **b**), but not in this complex trapped at the E/F interface are shown as red dashed lines.

used in structural studies, corroborated by computational investigations, as a mimic for phosphoryl groups in a variety of different enzymes (for a recent review see ref. [7]). Although ATP analogues such as ADP:AlF$_4^-$ represent non-physiological mimics of ATP hydrolysis, their use provides static snapshots of protein states approximately on the reaction coordinate inaccessible by other approaches.

In our work, EPR experiments allow the localization of the metal ion co-factor with respect to the ADP:AlF$_4^-$ unit. In the transition state of ATP hydrolysis for *Hp*DnaB, the Mn$^{2+}$ ion is in spatial proximity to the β-phosphate group of ADP as well as the AlF$_4^-$ unit (Fig. 7a) as concluded from the large $^{31}$P hyperfine coupling constant to the Pβ (a significantly smaller one is found for the Pα atom) and $^{19}$F and $^{27}$Al resonances observed in EDNMR, respectively. The structures of the only SF4-type helicases solved crystallographically, namely the *Bst*DnaB:GDP:AlF$_4^-$:DNA and *Aa*DnaB:ADP complexes (*Acquifex aeolicus*, PDB 4NMN[22]), support the finding of a Mn$^{2+}$ coordination to the β-phosphate group as also supported by the DFT calculations of the $^{31}$P hyperfine tensors revealing the same trends as observed experimentally (Supplementary Table 1, Supplementary Fig. 2). A similar experimental observation by EPR has been made for DbpA RNA helicase in complex with ADP[86].

Hydrogen bonds were identified spectroscopically by combining the information of high-frequency shifted proton resonances, spatial proximities probed in hPH correlation experiments and proton chemical-shift temperature coefficients (see Table 1 for a summary). Similar to other P-loop NTPases, in the *Hp*DnaB transition state trapped by solid-state NMR, residues S206, G208, K209, T210 and S211 of the Walker A motif were identified in coordinating the ADP phosphate groups by their backbone amino groups and by the sidechain of K209 yielding a dense hydrogen-bond network (see Fig. 7a for a schematic representation). DnaB helicases are characterized by a unique ARP[G/S]xGK[T/S] sequence of the Walker A motif with an Ala residue

instead of Gly in the first position[16]. Homologous residues were found to coordinate the phosphate chain in the crystal structure of DnaB from *Bacillus stearothermophilus* (currently *Geobacillus stearothermophilus*) crystallized with Ca$^{2+}$:GDP:AlF$_4^-$ and DNA (PDB accession code 4ESV[21], see Fig. 7b and Supplementary Table 2 for the averaged distances to the oxygen atoms of the phosphate groups). An important difference is the only partial occupation of NBDs in *Bst*DnaB with the transition-state analogue, whereas for *Hp*DnaB all binding sites are occupied[57] and highly symmetric as revealed by the absence of evident peak splitting in the hPH spectra.

The hPH spectrum reveals two key contacts in DNA recognition by DnaB, namely the coordination of D374 and G376 to the two structurally distinct DNA phosphate groups P1 and P2. The de-shielded proton resonances in combination with the almost vanishing temperature coefficient found for D374 point to an engagement of these two protons in hydrogen bonding (Figs. 3 and 4). In previous studies, we have also identified the sidechain of K373 in forming a salt-bridge to P2[44], which is also supported by the hPH spectrum showing a correlation of the resonance of the DNA phosphate group P2 to the K373 sidechain (Fig. 3c). These contacts are identical to those found in the crystal structure of the *Bst*DnaB:DNA complex (backbone amide proton of E382 and G384 and the sidechain of R381)[21] thus revealing similarities in DNA recognition for these two SF4-type helicases. The protein proton resonances contacting the DNA are not broadened or even split into several peaks indicating that all six DnaB subunits engage the DNA in a highly similar way, pointing to a closed hexamer rather than an extended open structure as observed for *Bst*DnaB (see Supplementary Fig. 11)[21]. This again agrees with our observation of a full saturation of all six NBDs with Mg$^{2+}$:ADP:AlF$_4^-$ therefore still indicating structural differences in the position of DnaB monomers in the *Hp*DnaB and *Bst*DnaB helicase complexes with transition-state analogues and DNA.

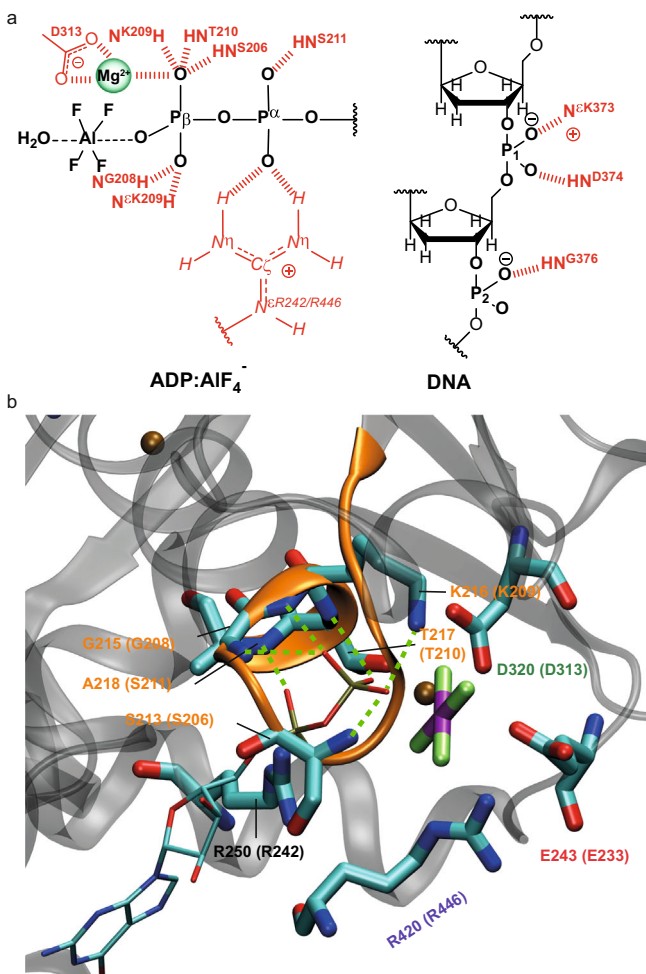

**Fig. 7 Comprehensive model for molecular recognition events involved in ADP and DNA binding to DnaB as obtained from the herein presented EPR/NMR results. a** Sketch of hydrogen-bond formation and spatial proximities as revealed by the hPH and chemical-shift temperature-dependence experiments for ADP and DNA coordination to $Hp$DnaB. The $Mg^{2+}$ co-factor has been placed in spatial proximity to the $AlF_4^-$ unit supported by the results from the EDNMR spectra (the coordinating aspartate located in the Walker B motif is shown additionally[113]). **b** Zoom into the nucleotide-binding domain for $Bst$DnaB:GDP:AlF$_4^-$:DNA (PDB accession code 4ESV). Residues given in brackets correspond to those in $Hp$DnaB. Green lines represent hydrogen bonds or spatial proximities as identified from the hPH correlation experiments.

An important feature revealed in our NMR analysis is the free rotational diffusion of the $AlF_4^-$ unit mimicking the departing phosphate group during ATP hydrolysis. The averaging of the $^{27}Al$ quadrupolar coupling constant in combination with the single $^{19}F$ resonance observed indicate that the $AlF_4^-$ unit (Fig. 5) is not coordinated tightly by the protein anymore, in contrast to the ADP for which we have observed a dense network of hydrogen bonds (Figs. 3 and 4). In contrast to the $Bst$DnaB structure, in the $Hp$DnaB complex studied herein ADP:AlF$_4^-$ moieties are present in all six catalytic pockets[57]. The observed uniformity, however, comes in contradiction with the sequential operation of catalytic subunits, as observed in several studied oligomeric P-loop ATPases[87,88]. Their subunits operate one after another so that the catalysis in one subunit is thermodynamically promoted by the substrate binding to the other subunit[89,90]. Hence, only one site stays at any moment in the conformation catalytically active for ATP hydrolysis.

The here reported free rotational diffusion of all six $AlF_4^-$ moieties within the tight hexamer of $Hp$DnaB (Fig. 5) could be explained in the following way (see also the extended discussion in Supplementary Note 1): The exergonic binding of the first ADP:AlF$_4^-$ moiety to a $Hp$DnaB subunit (subunit 1) brings it into its catalytically active, DNA-bound configuration with the Arg (R446) and Lys (K444) fingers of the adjoining subunit 2 interacting with the ADP:AlF$_4^-$:H$_2$O$_{cat}$ complex in the "catalytic" position (Fig. 6a). This suggestion is supported by our earlier observation that ADP:AlF$_4^-$ binding alone induces protein conformational changes and preconfigures the protein for DNA binding[57]. It is not clear yet for any of the P-loop NTPases how the stimulating moiety/moieties accelerate the hydrolysis. Binding of the ADP:AlF$_4^-$:H$_2$O$_{cat}$ to the subunit 2 transforms it in a similar way and, simultaneously, provides free energy for pulling the γ-phosphate-mimicking $AlF_4^-$ out of its catalytic position in the subunit 1—by K444 and R446 fingers of subunit 2—into one of the late transition-state positions as seen in the $Bst$DnaB structure, see Fig. 6b, c and ref. [21]. After this sequence of events repeats six times, all six protein subunits are in the same catalytic configuration being tightly fixed on the DNA strand (as revealed by the identified hydrogen bonds formed by D374 and G376 to the DNA phosphate groups, Fig. 3c) whereas their six $AlF_4^-$ moieties are, most likely, in positions similar to those taken by $AlF_4^-$ moieties in two of six catalytic sites of $Bst$DnaB, namely those on the subunit interfaces B/C and F/A, see Fig. 6c, Supplementary Fig. 8 and ref. [21]. In this state, the $AlF_4^-$ moieties are detached both from the ADP moiety and the Arg and Lys fingers. Structures of myosin with $H_2PO_4^-$, the physiological product of ATP hydrolysis, which is released in the final reaction step, in compatible positions are described in the literature[91]. Hence, we

**Table 1 Summary of spectroscopically identified hydrogen bonds.**

| H-Bond | $\delta(^1H^N) > 9$ ppm[a] | Correlation peak visible in hPH[b] | $\Delta\delta(H^N)/\Delta T > -4.6$ ppb/K[c] | $d$(N–O) < 3.5 Å[d] |
|---|---|---|---|---|
| S206 | Yes (9.4 ppm) | Yes Pβ (ADP) | n.d. | Yes (S213) |
| K209 | Yes (11.0 ppm) | Yes Pβ (ADP)[e] | Yes (0.0 ppb/K) | Yes (K216) |
| T210 | Yes (9.7 ppm) | Yes Pβ (ADP) | Yes (−0.9 ppb/K) | Yes (T210) |
| S211 | No (7.7 ppm) | Yes S211 Pα/Pβ (ADP), weak | n.d. | Yes (A218) |
| D371 | No (8.1 ppm) | Yes P1/P2 (DNA), weak | n.d. | No (D379) |
| K373 | No (8.5 ppm) | Yes P1/P2 (DNA)[e], weak | n.d. | Yes (R381[e]) |
| D374 | Yes (10.1 ppm) | Yes P1 (DNA) | Yes (0.0 ppb/K) | Yes (E382) |
| S375 | No (7.2 ppm) | Yes P1 (DNA), weak | n.d. | No (S383) |
| G376 | Yes (9.3 ppm) | Yes P2 (DNA) | Yes (−0.7 ppb/K) | Yes (G384) |

[a]Chemical-shift values taken from ref. [44].
[b]hPH spectra provide information about the phosphorous in close proximity to the protein proton.
[c]Missing data (n.d. not determined) can be mostly accounted to overlap in the 2D hNH spectra.
[d]Based on the $Bst$DnaB:DNA crystal structure PDB 4ESV, see also Supplementary Table 2.
[e]Besides backbone, also sidechain correlations are detected.

suggest that the mobile $AlF_4^-$ moiety in *Hp*DnaB mimics the phosphate group during a late transition state of ATP hydrolysis by *Hp*DnaB. To which extent the trapped transition state using a metal fluoride resembles the physiological transition state remains an open question at this stage.

Our results demonstrate that magnetic resonance is highly suitable to obtain structural and dynamic insights into the transition state of ATP hydrolysis of a bacterial DnaB helicase trapped by aluminum fluoride allowing a more profound understanding of the functioning of such complex motor proteins. EPR reveals the coordination of the metal ion co-factor to the β-phosphate group of ADP as well as to the $AlF_4^-$ unit, whereas proton-detected hPH solid-state NMR experiments combined with temperature dependences of proton chemical-shift values allow for identifying hydrogen bonds, which are crucial for the molecular recognition process of ADP and DNA binding to the DnaB helicase. NMR is one of the most sensitive techniques in proving hydrogen bonding with the additional advantage of shedding light onto dynamic processes, herein the free rotational diffusion of the $AlF_4^-$ unit mimicking the phosphate group transferred during ATP hydrolysis.

## Methods

**Sample preparation. Protein expression and purification**. The protein was cloned into the vector pACYC-duet1 (using the forward primer 5'-agtca-tatggatcatttaaagcatttgcag-3' containing a NdeI restriction site and reverse primer 5'-atactcgagttcaagttgtaactatatcataatcc-3' containing a XhoI site), and expressed in the E. coli strain BL21 Star (DE3) (One Shot® BL21 Star™ (DE3) Chemically Competent E. coli, Invitrogen™)[53]. Natural abundance and $^{13}C-^{15}N$ labeled *Hp*DnaB was prepared in buffer A (2.5 mM sodium phosphate, pH 7.5, 130 mM NaCl) as described in ref. [53]. In short, DnaB was recombinantly expressed in presence of $^{13}C$-glucose (2 g/L) and $^{15}N$-ammonium chloride (2 g/L) as sole sources of carbon-13 and nitrogen-15. In case of the deuterated protein, the protein was expressed in $D_2O$ in presence of deuterated $^{13}C$-glucose. The back-exchange was achieved by purifying the protein in a protonated buffer (2.5 mM sodium phosphate, pH 7.5, 130 mM NaCl).

**NMR sample preparation**. 0.3 mM *Hp*DnaB in buffer A was mixed with 5 mM $MgCl_2 \cdot 6H_2O$ and consecutively 6 mM of an $NH_4AlF_4$ solution (prepared by incubating 1 M $AlCl_3$ solution with a five-fold excess of 1 M $NH_4F$ solution (compared to $AlCl_3$) for 5 min) and 5 mM ADP and incubated for 2 h at 4 °C. Under these conditions a full occupation of binding sites has been observed[57]. 1 mM of $(dT)_{20}$ (purchased from Microsynth) was added to the complexes and reacted for 30 min at room temperature. The protein solution was sedimented[53,54,92] into the MAS-NMR rotor (16 h at 4 °C at 210,000 × g) using home-built tools[93]. In case of the DnaB:ADP:$AlF_4^-$ complex the DNA addition step was omitted.

**EPR sample preparation**. For EPR experiments, natural abundance DnaB was concentrated to 48 mg/ml (850 μM) using a Vivaspin 500 centrifugal filter with a cut-off of 30 kDa. The concentrated protein was incubated in presence of 6 mM ADP, 170 μM $Mn^{2+}$ and 7 mM $NH_4AlF_4$ for 2 h at 4 °C. After 2 h, glycerol was added to a concentration of 20%. The final concentrations were: DnaB 690 μM, ADP 5 mM, $Mn^{2+}$ 138 μM and $NH_4AlF_4$ 6 mM. An identical protocol was used for experiments performed on $^{13}C/^{15}N$ labeled DnaB.

**Solid-state NMR experiments**. Solid-state NMR spectra were acquired at 11.7, 14.1 and 20.0 T static magnetic-field strengths using an in-house modified Bruker 3.2 mm ($^{19}F$ and $^{27}Al$ NMR) probe and a 0.7 mm ($^1H$ NMR) triple-resonance ($^1H/^{31}P/^{13}C$) probe. The MAS frequencies were set to 17 and 100/105 kHz, respectively. The 2D spectra were processed with the software TOPSPIN (version 3.5, Bruker Biospin) with a shifted (2.0 or 3.0) squared cosine apodization function and automated baseline correction in the indirect and direct dimensions. For $^1H$-detected experiments, the sample temperature was set to 293 K[93] and varied in the range of 294–302 K for the temperature-dependence studies. A fast adjustment of the temperature in the bore of the magnet (typically causing $B_0$ instabilities) was achieved by a bore heating system implemented by the instrument manufacturer. This is crucial for detecting the rather small temperature dependences of proton chemical-shift values (on the order of several ppb/K)[62]. For $^{19}F$ (recorded at 14.1 T) and $^{27}Al$ (recorded at 11.7 T) MAS-NMR experiments, the sample temperature was adjusted to 278 K. $^1H$ and $^{31}P$-detected spectra were analysed with the software CcpNmr (version 2.4.2)[94–96] and referenced to 4,4-dimethyl-4-silapentane-1-sulfonic acid (DSS). $^{19}F$ and $^{27}Al$ spectra were referenced to internal standards. For more detail see the Source Data file. All samples were measured repeatedly. The samples were at least prepared twice and yield identical NMR spectra. The

temperature chemical-shift gradients were analysed with MATLAB, version 9.6.0 (R2019a).

**EPR experiments**. All experiments were conducted on a Bruker Elexsys E680 EPR spectrometer (Bruker Biospin) operating at W-band frequencies (~94.2 GHz). ENDOR measurements used a 250 W radiofrequency (rf) amplifier. The temperature was generally set to 10 K.

Electron–electron double resonance (ELDOR)-detected NMR spectra were acquired with a shot repetition time of 1 ms and the echo-detected hole-burning sequence $t_{HTA}$—T—$t_p$ – τ – $2t_p$ – τ—echo, with $t_{HTA} = 50$ μs, T = 10 μs, $t_p = 100$ ns, τ = 1400 ns and an integration window of 1400 ns. The frequency of the high-turning angle (HTA) pulse was incremented in steps of 0.1 MHz over the measured range. A+/− phase cycle on the first π/2 pulse of the echo was used to eliminate unwanted coherence transfer pathways. The power of the HTA pulse, generated by the ELDOR channel of the spectrometer, was optimized such that the observed lines were as intense as possible without being broadened by saturation effects. The nutation frequency $v_1$ at the centre of the resonator was ca. 6 MHz or ca. 12 MHz, denoted as low and high HTA pulse power, respectively. The settings were held constant between protein samples and the corresponding control samples. Yet it is important to note that exact reproducibility of peak intensities between runs may be difficult with the resonator used because the resonator profile strongly affects line intensities in EDNMR, and hence a careful experimental setup is required.

Davies ENDOR spectra were acquired with a shot repetition time of 5 ms and with the sequence $t_{inv}$—T—$t_p$ – τ – $2t_p$ – τ—echo, where during the time T, an rf pulse was applied. The inversion pulse length was set to 200 ns, and the rf pulse length to 50 μs. The echo was integrated symmetrically around the echo maximum over a time of 400 ns. Due to enormous time overhead on this particular spectrometer, we did not use stochastic acquisition mode and used 10 shots per point.

Mims ENDOR spectra were acquired with a shot repetition time of 2.5 ms and with the sequence $t_p$ – τ –$t_p$—T—$t_p$ – τ—echo, where during the time T, an rf pulse was applied. The interpulse delay τ was set to 1200 ns, corresponding to the phase memory time $T_m$, where detection of small hyperfine couplings is most sensitive[97], and the rf pulse length to 25 μs.

Raw EDNMR data were background corrected with a Lorentzian line that was fitted to the central hole, and normalized to the signal intensity far off-resonance, i.e. the peak intensity corresponds to the relative hole depth. Fitting of the EPR spectra was performed with EasySpin (version 6.0.0).

**DFT calculations of hyperfine tensors**. DFT calculations were performed on small clusters mimicking the coordination sphere of the metal ion co-factor extracted from the PDB structures (*Bst*DnaB: accession code 4ESV and *Aa*DnaB: accession code 4NMN, see Supplementary Fig. 2). Hydrogen atoms were added to saturate terminating groups and their positions were optimized on a TPSS[98]/def2-SVP[99] level using TURBOMOLE (version 6.0)[100,101]. In all TURBOMOLE SCF calculations, an energy convergence criterion of $10^{-7}$ $E_h$ and in all geometry optimizations an energy convergence criterion of $5 \times 10^{-7}$ $E_h$ was chosen. The integration grid was set to m4 and the RI approximation was used. Hyperfine coupling tensors were calculated in the ADF suite (version 2013)[102] on a B3LYP[103,104]/TZ2P[105] level of theory. The INTEGRATION keyword was set to 6.0 and in the SCF calculation an energy convergence criterion of $10^{-6}$ $E_h$ was used.

**Reporting summary**. Further information on research design is available in the Nature Research Reporting Summary linked to this article.

## Data availability

The NMR and EPR spectra can be accessed at https://doi.org/10.3929/ethz-b-000501034. The following PDB structures were used in this study: 4ZC0, 4NMN, 4ESV, 3CMW and 3PUW. All experimental NMR parameters are provided as a Source Data file. Protein resonance assignments are available from the BMRB database (www.bmrb.wisc.edu, accession code 27879). Source data are provided with this paper.

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

## Acknowledgements

This work was supported by the ETH Career SEED-69 16-1 (T.W.) and the ETH Research Grant ETH-43 17-2 (T.W.), the Deutsche Forschungsgemeinschaft (DFG, German Research Foundation, project number 455240421 and Heisenberg fellowship, project number 455238107, T.W.), an ERC Advanced Grant (B.H.M., grant number 741863, Faster) and by the Swiss National Science Foundation (B.H.M., grant number 200020_159707 and 200020-188711), the German Academic Exchange Service (DAAD, A.Y.M.) and the EvoCell Program of the Osnabrueck University (M.I.K.). T.W. acknowledges discussions with Prof. Matthias Ernst and Dr. Denis Lacabanne.

## Author contributions

A.A.M., L.A.V. and T.W. performed the NMR experiments, N.W. and D.K. the EPR experiments. R.C. prepared the samples. A.D. modified the NMR probes for ²⁷Al and ¹⁹F experiments. T.W. performed the DFT calculations. M.I.K. and A.Y.M. performed the structural modellings. A.A.M., L.A.V., N.W., D.K., M.E.W., J.Z., H.E., G.J., A.Y.M., A.B., B.H.M. and T.W. analyzed the data. D.K., A.Y.M., B.H.M. and T.W. designed and supervised the research. All authors contributed to the writing of the manuscript.

## Competing interests

The authors declare no competing interests.
