## [Peer Review File · Nature Communications]

Spectroscopic glimpses of the transition state of ATP hydrolysis trapped in a bacterial DnaB helicaseREVIEWER COMMENTS

Reviewer #1 (Remarks to the Author):

This manuscript describes combined EPR and NMR methods applied to characterize the ATP hydrolysis transition state of the bacterial DnaB helicase. The transition state is mimicked by using the well established $Mg^{2+}/ADP/AIF^{-4-}$ co factor. This work is an extension of an earlier, extensive work published in this journal (ref 66) in 2019, now targeting the mimicked transition state in more details with focus on the AIF^{-4-} coordination site its dynamics in the presence and absence of bound DNA. The main findings are the identification of hydrogen bonds to the $Mg^{2+}/ADP/AIF^{-4-}$ co factor obtained by fast 1H MAS and the highly mobile character of AIF^{-4-} . The EPR measurements give some information regarding the local environment of the metal co-factor achieved by the well known approach of substituting the diamagnetic Mg^{2+} with Mn^{2+} . This work uses elegant, state of the art magnetic resonance methodology and adds new understanding regarding the ATP hydrolysis mechanism. I recommend publication of a revised version where the following issues (some crucial) should be addressed.

1. I miss the proof that the substitution of Mg^{2+} with Mn^{2+} retains the ATP hydrolysis activity. Please include such activity tests in the SI. Indeed, this has been shown to be the case for other systems but has to be proven for this one as well.
2. Important : Fig. 6 a show the direct coordination of the Mg^{2+} to ^{19}F saying “in agreement with the EDNMR results”. This is not really correct, the EDNMR spectrum shows a weak single ^{19}F peak and the only conclusion that one may draw from this is that the ^{19}F is in the vicinity of the Mn^{2+} . Direct coordination should yield a significant anisotropic hyperfine coupling that should be easily observed by ENDOR. ^{19}F is a very friendly nucleus, like 1H , and ENDOR should be straight forward. The $Mn-F$ distance can be obtained from the anisotropic hyperfine coupling.
3. The dissociation constant of $Mg^{2+}/ADP/AIF^{-4-}$ is often low and I guess that this is this is so also for the present case as the authors took the protein in excess to minimize the amount of free $Mn^{2+}/ADP/AIF^{-4-}$. Is the dissociation constant known? this should be discussed and explained in the main text.
4. For the NMR experiments $Mg^{2+}/ADP/AIF^{-4-}$ is used in high excess to ensure that all proteins have a $Mg^{2+}/ADP/AIF^{-4-}$ bound. This means that there is a lot of free $Mg^{2+}/ADP/AIF^{-4-}$. This, however, does not seem to create a problem and the intensity of free AIF^{-4-} in the spectra is indeed small. Is this because of the sedimentation? Please explain in the text.
5. The authors do not mention any evidence that the $Mn^{2+}/ADP/AIF^{-4-}$ is indeed bound to the protein. They do not show any interaction with protein nuclei as shown in ref 59 for example. I think that the evidence comes from the different ^{31}P ENDOR which shows a weakly bound ^{31}P in the presence of the protein and the shift of the Mn^{2+} signal, meaning a change in its hyperfine coupling and coordination mode (Fig. 1a). This is important and should be added.
6. What is the broad pick around 130 MHz in the purple spectrum in Fig. 1A.
7. What does it mean that parts of the cyan spectrum are reproduced from reference ^{31}P ENDOR spectra?
8. Why were the NMR experiments carried out on samples in the presence and absence of DNA and the EPR experiments only in the absence of DNA.
9. Are the two ^{31}P chemical shifts observed for ADP consistent with the proximity to the Mg^{2+} as observed by EPR?
10. There is an extensive ENDOR study by Sun Un on Mn^{2+} binding to nucleotides and their associated ^{31}P hyperfine couplings and DFT calculations. Will be nice to mention this work.
11. In the first sentence of the discussion the authors wrote “ EPR experiments allow the localization of the metal co-factor within the NBD”. This sentence is misleading, all the interactions of the Mn^{2+} are within the ADP/AIF^{-4-} unit, so there is no information regarding the location within the protein, just that it is bound.
12. In p.11 (would be nice to have page numbers..) the authors refer to a fast exchange process, please explain – what type of exchange process.
13. Will be helpful for the reader to have final schematic figure that shows the ATP hydrolysis mechanism in the light of the new insights obtained from this work.

Reviewer #2 (Remarks to the Author):

This manuscript reports on an investigation of a ADP aluminum fluoride binding to bacterial DnaB helicase from *H. pylori*, by solid-state NMR spectroscopy. ^1H detection with fast MAS was used to delineate the protein residues participating in the interactions with ADP and DNA; temperature-dependent ^1H chemical shifts were used as indicators of residues forming hydrogen bonds; these studies were supplemented by ^{19}F and ^{27}Al spectra to infer the dynamics of the AlF_4^- ion in the context of the complex with the protein. Overall, this work is excellent from the technical standpoint, the spectral resolution is high, proper controls have been carried out, and the conclusions based on the experimental results are generally sound, at least when it concerns interpretation of spectroscopic data.

This said, when judging the suitability for publication of a study in interdisciplinary journals, such as *Nat. Comms.*, one has to consider and weight two factors: i) new insights into the system under investigation, or/and ii) methodological innovation. In my opinion, the current report does not meet the stringent criteria for publication in *Nat. Comms.*, and my recommendation is "publish elsewhere". Specific points are below.

Biological relevance. The most interesting finding of the work is the identification of the residues involved in hydrogen bonding, on the basis of the temperature-dependent chemical shifts. This is typically not possible with most solid-state NMR instruments due to temperature fluctuations. This finding alone, however, is not sufficient to warrant publication in *Nat. Comms.*

A major portion of the manuscript, as written, concerns the characterization of aluminum fluoride in the protein complex. While the use of metal fluorides as mimics of intermediate states of ATP hydrolysis is common for structural characterizations, particularly by X-ray crystallography (and the authors use this as a justification for their work), these species have different electronic properties from the bona fide ATP hydrolysis intermediates. The discovery by the authors that the AlF_4^- is rotating when bound to DnaB, while interesting, may or may not be of relevance to the ATP hydrolysis mechanism. This part of the manuscript and the corresponding discussion have to be revised to make it clear to the reader that the mechanistic hypotheses they put forth are not based on the actual biologically relevant state.

Technical innovation. All spectroscopy, while state of the art, is well established, by the authors or by others.

Reviewer #3 (Remarks to the Author):

In this manuscript, Malär et al. investigate the bacterial helicase DnaB from *Helicobacter pylori* by solid-state NMR spectroscopy and electron paramagnetic resonance. The transition state of this ATPase is mimicked with $\text{ADP}\cdot\text{AlF}_4^-$. The authors determine structural details of the transition state by exploiting NMR active nuclei: ^{19}F , ^{27}Al , ^{31}P . Furthermore, they investigate the positioning of the Mg^{2+} ion and the surrounding amino acid side chains of the protein. The findings are interpreted in the context of published crystal structures of close homologues of DnaB from *(Geo)Bacillus stearothermophilus* and *Acquifex aeolicus* in the presence of analogues mimicking the respective transition states.

Relation to established literature. The manuscript builds on previous work by the authors, in which the enzymatic process of ATP hydrolysis in DnaB was investigated (Wiegand et al. *Nat. Commun.* 2019) and the authors presented NMR spectra of the transition state. The methodology employed herein has been established in previous works from the authors: paramagnetic substitution of the Mg^{2+} cofactor (Wiegand et al. *Angew. Chem.* 2017), experiments for probing nucleotide (Wiegand et al. *ChemBioChem* 2019) and DNA binding (Wiegand et al. *Angew. Chem.* 2016), hydrogen bonds identified via temperature series (Mälär et al. *ChemRxiv* 2021), localization of the metal ion cofactor from pseudo contact shifts (Zehnder et al. *Chem. Eur. J* 2021). The present manuscript constitutes a well-written synopsis and extension of these efforts.

Given the authors' extensive publication record of solid-state NMR applied to DnaB, they need to

enunciate more clearly what is the novel contribution of this particular manuscript that would appeal to the broad readership of Nature Communications.

Significance. The authors showcase a wide array of magnetic resonance methods applied to the DnaB model ATPase. As such, this work is of interest in particular to the solid-state NMR and EPR communities. Taking an inventory of all the heteronuclei that occur in the transition state analogue, the study is a very solid contribution to the field, albeit not radically new.

In terms of structural biology, the manuscript validates published crystal structures of related homologue DnaB proteins and adds a few novel details that are inaccessible to crystallographic studies.

Noteworthy results. As a new feature of the transition state, the authors discover that the AIF₄⁻ ion, which is thought to mimic the trigonal-bipyramidal state of the phosphoryl transfer reaction, undergoes fast reorientation.

Contacts between the protein, specifically the Walker A motif, and ADP as well as the phosphate backbone of the DNA substrate are identified.

The documentation of the methods is sufficient for a specialized readership and the conclusions of the authors are mostly supported by the data.

Major points:

1. The crystal structures of DnaB homologues and other P-loop ATPases form the basis for interpreting the NMR data: assignment of cross peaks, DFT calculations. Given the central importance of these structures for the understanding of this study, it would be important that the authors clearly state in the introduction what structures have been previously published and how they compare to and possibly differ from the transition state of HpDnaB. This would allow the reader to judge where it is appropriate to draw analogies and mingle structural analysis.

2. For example, the DFT calculations are based on crystal structures. The authors draw the conclusion that the coordination mode of the Mn²⁺ ion must be similar in HpDnaB. Could the authors identify and test alternative structures in which this ion sits closer to the alpha-phosphate? If there are no examples of such alternative geometries, one might argue that the question answered was not a pressing one in the first place.

3. The authors employ three methods to detect hydrogen bonds: chemical shifts, hPH correlation experiments and temperature coefficients of chemical shifts. A table listing each putative hydrogen bond, which methods can detect it, and whether the hydrogen bond is compatible with crystal structure(s) would be handy for the reader.

4. The authors find that the AIF₄ rotates rapidly and is not rigidified by coordinating side chains. On the other hand, the moiety is immobilized to such an extent that crystallization is feasible and NMR samples are stable for days if not weeks. Could the authors comment on the stability of the complex and on motions other than rotational diffusion of the AIF₄ unit?

5. The ¹⁹F chemical shifts of AIF₄⁻ in the absence and presence of DNA in Figure 4 are not identical while the shifts of the free species are well reproducible. Could the authors comment on this difference?

6. In Figure 5, the authors speculate that snapshots obtained from multiple crystal structures of different P-loop NTPases retrace a series of reaction steps, from early to late transition state. Could the authors provide concrete evidence or test this hypothesis?

For example: In Figure 6, the authors project their model of the nucleotide-binding domain of HpDnaB onto the crystal structure of BstDnaB. Could this model be validated with point mutants of critical residues, for example the activating lysine and arginine residues, in order to manipulate the reaction? Another possibility would be to align the sequences of the homologues, identify critical amino-acid substitutions and introduce these into HpDnaB. If the hypothesis of the authors holds, one would expect that the mobility of the AIF₄ moiety could be tuned in this manner. As a last resort, the authors could simply perform ¹⁹F and ²⁷Al NMR spectroscopy on trapped transition states of those homologues that they predict feature earlier or later transition states. One would expect to observe a difference in the mobility of AIF₄.

We thank the referees for carefully reading the manuscript, the overall positive feedback and the constructive comments which we have taken into account for the revised manuscript. A point-by-point answer is given below and relevant changes are marked in yellow in the revised manuscript.

We are looking forward to the final decision.

Yours sincerely,

Thomas Wiegand

Reviewer #1 (Remarks to the Author):

This manuscript describes combined EPR and NMR methods applied to characterize the ATP hydrolysis transition state of the bacterial DnaB helicase. The transition state is mimicked by using the well established $Mg^{2+}/ADP/AlF_4^-$ co factor. This work is an extension of an earlier , extensive work published in this journal (ref 66) in 2019, now targeting the mimicked transition state in more details with focus on the AlF_4^- coordination site its dynamics in the presence and absence of bound DNA. The main findings are the identification of hydrogen bonds to the $Mg^{2+}/ADP/AlF_4^-$ co factor obtained by fast 1H MAS and the highly mobile character of AlF_4^- . The EPR measurements give some information regarding the local environment of the metal co-factor achieved by the well known approach of substituting the diamagnetic Mg^{2+} with Mn^{2+} . This work uses elegant, state of the art magnetic resonance methodology and adds new understanding regarding the ATP hydrolysis mechanism. I

recommend publication of a revised version where the following issues (some crucial) should be addressed.

1. I miss the proof that the substitution of Mg^{2+} with Mn^{2+} retains the ATP hydrolysis activity. Please include such activity tests in the SI. Indeed, this has been shown to be the case for other systems but has to be proven for this one as well.

We agree with the reviewer that such activity tests are crucial when using Mn^{2+} instead of Mg^{2+} for structural investigations. However, those experiments were already performed for *HpDnaB* (see reference 30, *Soni, R.K., Mehra, P., Choudhury, N.R., Mukhopadhyay, G. & Dhar, S.K. Nucleic Acids Res. 31, 6828-6840 (2003).*, already cited in our manuscript). In this work the authors show that the DNA unwinding capability (which is driven by ATP-hydrolysis) is maintained with Mn^{2+} (around 80% of the activity found for Mg^{2+}). We have clarified our statement in the manuscript.

“The native Mg^{2+} cofactor is replaced for such studies by the EPR-observable paramagnetic Mn^{2+} analogue²⁹ (the biological functionality is maintained to about 80% under such conditions compared to the one observed in presence of Mg^{2+} ³⁰).

2. Important : Fig. 6 a show the direct coordination of the Mg^{2+} to ^{19}F saying “in agreement with the EDNMR results”. This is not really correct, the EDNMR spectrum shows a weak single ^{19}F peak and the only conclusion that one may draw from this is that the ^{19}F is in the vicinity of the Mn^{2+} . Direct coordination should yield a significant anisotropic hyperfine coupling that should be easily observed by ENDOR. ^{19}F is a very friendly nucleus , like 1H , and ENDOR should be straight forward. The Mn-F distance can be obtained from the anisotropic hyperfine coupling.

We thank the reviewer for this important remark. Figure 6 has been updated to reflect both spatial proximity and chemical bonding and the suggestion of a direct coordination to AlF_4^- has been removed, i.e. we have modified Figure 6a as well as our statement in the manuscript.

“The Mg^{2+} cofactor has been placed in spatial proximity to the AlF_4^- unit supported by the results from the EDNMR spectra.” (Legend Figure 6)

This proximity of ^{19}F to Mn^{2+} is in agreement with the ^{19}F weak coupling observed by EDNMR. Further, we added an ENDOR spectrum measured in Q band to the SI (Figure S1), where a weak ^{19}F coupling is observed in agreement with the EDNMR spectrum. As the reviewer pointed out, in principle from the anisotropic part of the hyperfine coupling the ^{19}F - Mn^{2+} distance could be inferred, here however resolution is too limited to distinguish between isotropic and anisotropic contributions to the hyperfine coupling, which is a prerequisite for determining the distance. The ENDOR spectrum is referred to in the main text stating, *“The weakly coupled ^{19}F is also observed by ENDOR (Figure S1) and indicates its proximity to Mn^{2+} .”*

3. The dissociation constant of $\text{Mg}^{2+}/\text{ADP}/\text{AlF}_4^-$ is often low and I guess that this is this is so also for the present case as the authors took the protein in excess to minimize the amount of free $\text{Mn}^{2+}/\text{ADP}/\text{AlF}_4^-$. Is the dissociation constant known? this should be discussed and explained in the main text.

The dissociation constant is not known and we have determined the excess of $\text{ADP}:\text{AlF}_4^-$ qualitatively by judging the NMR spectra regarding full occupation of the binding sites. We have mentioned that in the manuscript. An excess poses typically no problems in the solid-state NMR experiments, since e.g. for ^{31}P NMR studies we can only select the immobilized, bound nucleotides by performing cross-polarization experiments, in which the mobile species cannot be detected.

“Under these conditions a full occupation of binding sites has been observed⁵⁷.”

4. For the NMR experiments $\text{Mg}^{2+}/\text{ADP}/\text{AlF}_4^-$ is used in high excess to ensure that all proteins have a $\text{Mg}^{2+}/\text{ADP}/\text{AlF}_4^-$ bound. This means that there is a lot of free $\text{Mg}^{2+}/\text{ADP}/\text{AlF}_4^-$. This, however, does not seem to create a problem and the intensity of free AlF_4^- in the spectra is indeed small. Is this because of the sedimentation? Please explain in the text.

Indeed, the reviewer is right that we can still detect free AlF_4^- and related species in solution which is particularly visible in the ^{19}F MAS spectra in Figure 4. In a sedimented sample, we still have around 50% of the rotor filled with buffer and thus mobile AlF_4^- species that we detect. However, as pointed out by the reviewer, this reduces the amount of free AlF_4^- detected in such spectra. We have clarified this in the manuscript.

“The additional sharp ^{19}F resonances visible in the spectra are attributed to the excess of AlF_4^- and related species present in the supernatant of the NMR rotor (roughly 50 weight percentages after sedimentation⁷⁹). Around 4% of the AlF_4^- remains in the supernatant after the rotor-filling step.”

5. The authors do not mention any evidence that the $\text{Mn}^{2+}/\text{ADP}/\text{AlF}_4^-$ is indeed bound to the protein. They do not show any interaction with protein nuclei as shown in ref 59 for example. I think that the evidence comes from the different ^{31}P ENDOR which shows a weakly bound ^{31}P in the presence of the protein and the shift of the Mn^{2+} signal, meaning a change in its hyperfine coupling and coordination mode (Fig. 1a). This is important and should be added.

We have extended the explanation in the main text, which now states,

“To corroborate that these resonances are due to DnaB-bound $Mn^{2+}:ADP:AlF_4^-$ and to rule out that these correlations originate from the formation of the $Mn^{2+}:ADP:AlF_4^-$ complex in solution, we recorded EDNMR spectra on a frozen control solution in the absence of protein and indeed we do not observe any ^{19}F and ^{27}Al resonances (purple spectrum in Figure 1a).”

Since in these experiments the protein has natural isotope abundance, there are no protein-specific resonances (such as ^{15}N or ^{13}C) to be evaluated. As can be expected, there are ^{14}N resonances visible in EDNMR below 30 MHz that are not shown because this region naturally suffers from strong peak overlap which reduces the attainable information.

6. What is the broad pick around 130 MHz in the purple spectrum in Fig. 1A.

The broad feature around 130 MHz in the purple spectrum in Figure 1a is a phase twist rather than a peak, and the origin of this is not perfectly clear, however this has also been observed previously at lower intensity in EDNMR spectra of Mn-ATP, notably in the absence of ^{19}F (see Figure 4a in Goldfarb's & Un's work (JMR 2018, DOI: <https://doi.org/10.1016/j.jmr.2018.07.007> or also weakly visible in Figure 4b of eMagRes, 2017, Vol 6: 101–114. DOI 10.1002/9780470034590.emrstm1516). Putatively, this is an effect due to the Mn hyperfine coupling and the intensity may depend strongly on pulse parameters (which differ between the different examples). We mention now in the legend of Figure 1 this possible explanation.

“ marks a broad feature putatively due to the Mn hyperfine coupling.”*

7. What does it mean that parts of the cyan spectrum are reproduced from reference ^{31}P ENDOR spectra ?

We have previously published the EDNMR spectrum of DnaB:ADP in another context and have taken the spectrum from reference 64 in which we have only shown a zoom into this spectrum. We have solved the confusion by rephrasing the sentence in the manuscript.

“The cyan spectrum is reproduced from reference⁶⁴.”

8. Why were the NMR experiments carried out on samples in the presence and absence of DNA and the EPR experiments only in the absence of DNA.

Previous solid-state NMR investigations have shown that the $ADP:AlF_4^-$ states are highly similar in presence and absence of DNA (see reference 66) and we thus decided to record the EPR spectra only on one of these complexes.

9. Are the two ^{31}P chemical shifts observed for ADP consistent with the proximity to the Mg^{2+} as observed by EPR?

The ^{31}P chemical-shift values are highly sensitive to both, the chemical environment, and thus Mg^{2+} coordination, as well as the phosphate backbone geometry and it is not possible to disentangle these effects. In the case of $ADP:AlF_4^-$ the most significant contribution to the ^{31}P shifts seems to arise from the AlF_4^- contribution which shifts the Pa resonance to higher and the Pb resonance to lower ppm-values compared to the ADP-bound state in agreement with observations in aluminium phosphate gels and glasses (see ref 86). It is therefore not possible to explicitly quantify the effect from Mg^{2+} coordination on the ^{31}P chemical shift.

10. There is an extensive ENDOR study by Sun Un on Mn^{2+} binding to nucleotides and their associated ^{31}P hyperfine couplings and DFT calculations. Will be nice to mention this work.

We have included this important reference in the manuscript.

“This value is similar to published values for an ADP:Mn²⁺ complex in which the Mn²⁺ ion binds symmetrically to the two ADP phosphate groups^{65,66} or an ATP:Mn²⁺ complex⁶⁷.”

11. In the first sentence of the discussion the authors wrote “ EPR experiments allow the localization of the metal co-factor within the NBD”. This sentence is misleading, all the interactions of the Mn²⁺ are within the ADP/AlF₄⁻ unit, so there is no information regarding the location within the protein, just that it is bound.

We agree with the reviewer and have rephrased the sentence.

“EPR experiments allow the localization of the metal ion co-factor within the ADP:AlF₄⁻ unit.”

12. In p.11 (would be nice to have page numbers..) the authors refer to a fast exchange process, please explain – what type of exchange process.

For a description of the rotation of the AlF₄⁻ unit we refer to the ²⁷Al MAS NMR section where we describe this in detail. However, we have also rephrased the mentioned sentence.

“Interestingly, only one ¹⁹F resonance line at around -146 ppm is detected for the AlF₄⁻ group pointing to a fast chemical-exchange process, most probably a rotation of the unit (vide infra, for the ¹⁹F spectrum in the absence of protein see Figure S7).”

13. Will be helpful for the reader to have final schematic figure that shows the ATP hydrolysis mechanism in the light of the new insights obtained from this work.

Since we only describe one snapshot of ATP hydrolysis, namely the transition state, in this work, it is impossible to conclude on the overall ATP hydrolysis reaction mechanism. Furthermore, as we explicitly write in the revised manuscript, the mechanism of ATP hydrolysis by P-loop has yet to be determined. Still, in the revised manuscript we provide in the Supplementary Section a detailed description of the mechanistic steps in the turnover of the best studied P/loop NTPases, such as F1/ATPase and myosine with respective references. The key point of our work is a detailed structural and dynamic characterization of the transition state trapped by ADP:AlF₄⁻, particularly by establishing solid-state NMR strategies to probe hydrogen bonding in a large protein-DNA complex. These findings are summarized in Figure 6 and we have included an additional Table 1 summarizing the spectroscopically identified hydrogen bonds to ADP and DNA (see also the suggestion of Reviewer 3). Figure 6 also illustrates the rotation of the AlF₄-unit we have characterized for the first time in detail, although it still remains unclear whether this is characteristic for ATP hydrolysis (and the late transition state trapped) or if it is related to the chemical nature of the ATP-analogue mimic. According to the comments of reviewer 2 we made this clearer in our article. We plan to investigate this effect further in other protein systems in the upcoming years.

Reviewer #2 (Remarks to the Author):

This manuscript reports on an investigation of a ADP aluminum fluoride binding to bacterial DnaB helicase from *H. pylori*, by solid-state NMR spectroscopy. ^1H detection with fast MAS was used to delineate the protein residues participating in the interactions with ADP and DNA; temperature-dependent ^1H chemical shifts were used as indicators of residues forming hydrogen bonds; these studies were supplemented by ^{19}F and ^{27}Al spectra to infer the dynamics of the AlF_4^- ion in the context of the complex with the protein. Overall, this work is excellent from the technical standpoint, the spectral resolution is high, proper controls have been carried out, and the conclusions based on the experimental results are generally sound, at least when it concerns interpretation of spectroscopic data.

This said, when judging the suitability for publication of a study in interdisciplinary journals, such as *Nat. Comms.*, one has to consider and weight two factors: i) new insights into the system under investigation, or/and ii) methodological innovation. In my opinion, the current report does not meet the stringent criteria for publication in *Nat. Comms.*, and my recommendation is "publish elsewhere". Specific points are below.

Biological relevance. The most interesting finding of the work is the identification of the residues involved in hydrogen bonding, on the basis of the temperature-dependent chemical shifts. This is typically not possible with most solid-state NMR instruments due to temperature fluctuations. This finding alone, however, is not sufficient to warrant publication in *Nat. Comms.*

A major portion of the manuscript, as written, concerns the characterization of aluminum fluoride in the protein complex. While the use of metal fluorides as mimics of intermediate states of ATP hydrolysis is common for structural characterizations, particularly by X-ray crystallography (and the authors use this as a justification for their work), these species have different electronic properties from the bona fide ATP hydrolysis intermediates. The discovery by the authors that the AlF_4^- is rotating when bound to DnaB, while interesting, may or may not be of relevance to the ATP hydrolysis mechanism. This part of the manuscript and the corresponding discussion have to be revised to make it clear to the reader that the mechanistic hypotheses they put forth are not based on the actual biologically relevant state.

We agree with the reviewer that AlF_4^- is not involved in the physiological ATP hydrolysis mechanism. However, metal fluorides have been used successfully (corroborated by computational work) in trapping transition states in phosphoryl-transferring enzymes. The transition state is in general inaccessible for structural studies and we believe that the state of DnaB trapped by AlF_4^- resembles the physiological state as close as possible. We have clarified this in our manuscript at the beginning and end of the discussion.

"The transition state of ATP hydrolysis in the bacterial DnaB helicase from *Helicobacter pylori* has been trapped by using the mimic $\text{ADP}:\text{AlF}_4^-$. Such metal fluorides have been successfully used in structural studies, corroborated by computational investigations, as a mimic for phosphoryl groups in a variety of different enzymes (for a recent review see reference 7). Although ATP analogues such as $\text{ADP}:\text{AlF}_4^-$ represent non-physiological mimics of ATP hydrolysis, their use allows structural insights into static snapshots of such protein states inaccessible by other approaches."

"To which extent the trapped transition state using a metal fluoride resembles the physiological transition state remains an open question at this stage."

Technical innovation. All spectroscopy, while state of the art, is well established, by the authors or by others.

We thank the reviewer for the overall positive comments and would like to stress that we have also established and used for this work a proton-detected ^{31}P , ^1H correlation experiment at 100 kHz MAS which gives us direct information about protons in very close proximity to the nucleotide phosphate groups. Note that this is the first hPH correlation spectrum reported at this spinning frequency to the best of our knowledge. Thanks to this technology it is possible to use hPH correlation spectroscopy for hydrogen-bond determination using a sub-milligram sample amount, which was so far not possible with any of the previous equipment. This is an important step for proving hydrogen bonding in protein-nucleic acid complexes by solid-state NMR and to derive nucleotide binding modes, even in quite large systems as the one we looked at and is in our opinion an important technological advancement of solid-state NMR. NMR benefits from the key advantage of probing the engagement of protons in hydrogen bonding which is nearly impossible by other state-of-the-art structure characterization techniques.

We have summarized the spectroscopic tools we have applied to identify hydrogen bonds now in the new Table 1 (see also the suggestion of Reviewer 3).

Reviewer #3 (Remarks to the Author):

In this manuscript, Malär et al. investigate the bacterial helicase DnaB from *Helicobacter pylori* by solid-state NMR spectroscopy and electron paramagnetic resonance. The transition state of this ATPase is mimicked with ADP.AIF₄⁻. The authors determine structural details of the transition state by exploiting NMR active nuclei: ^{19}F , ^{27}Al , ^{31}P . Furthermore, they investigate the positioning of the Mg²⁺ ion and the surrounding amino acid side chains of the protein. The findings are interpreted in the context of published crystal structures of close homologues of DnaB from (Geo)Bacillus stearothermophilus and Acquifex aeolicus in the presence of analogues mimicking the respective transition states.

Relation to established literature. The manuscript builds on previous work by the authors, in which the enzymatic process of ATP hydrolysis in DnaB was investigated (Wiegand et al. Nat. Commun. 2019) and the authors presented NMR spectra of the transition state. The methodology employed herein has been established in previous works from the authors: paramagnetic substitution of the Mg²⁺ cofactor (Wiegand et al. Angew. Chem. 2017), experiments for probing nucleotide (Wiegand et al. ChemBioChem 2019) and DNA binding (Wiegand et al. Angew. Chem. 2016), hydrogen bonds identified via temperature series (Mälär et al. ChemRxiv 2021), localization of the metal ion cofactor from pseudo contact shifts (Zehnder et al. Chem. Eur. J 2021). The present manuscript constitutes a well-written synopsis and extension of these efforts.

Given the authors' extensive publication record of solid-state NMR applied to DnaB, they need to enunciate more clearly what is the novel contribution of this particular manuscript that would appeal to the broad readership of Nature Communications.

We thank the reviewer for the advice and have clarified in our article what the novel contribution of our work is. Besides the observation of a highly mobile AIF₄⁻ unit, not characterized in detail before, we have developed and applied a strategy to probe hydrogen bonding by solid-state NMR using proton chemical-shift temperature gradients (which we illustrate here for the first time to a "real" protein system, namely a large protein-DNA complex of significant biological relevance) and, even more importantly, the first hPH correlation spectrum reported beyond 100 kHz MAS using sub-milligram protein amounts. As pointed out by reviewer 2, an interesting finding of the work is

the identification of the residues involved in hydrogen bonding, on the basis of the temperature-dependent chemical shifts. This is typically not possible with most solid-state NMR instruments due to temperature fluctuations. The herein presented shows the first application of this methodology to a particularly large protein complex, illustrating its robust applicability and great strength in elucidating hydrogen networks for biologically very important questions as the nucleotide binding modes. The reported hPH spectra only show the protons in very close proximity to the DNA or ADP phosphate groups thus yielding highly specific distance restraints that will enter structure calculation in the next future. The herein presented methodology can be widely extended to any other protein-DNA/RNA complexes, such as primases, viruses, etc. motivating its interest for a broad readership. We have added the following sentences to our manuscript.

“A further advantage is the sensitivity of NMR in identifying hydrogen bonds, which is often not achievable by standard structure determination techniques, such as X-ray crystallography or cryo-electron microscopy, in which resolution in the order of 1 Å (for cryo-EM a slightly lower resolution might be sufficient⁵⁶) has to be achieved. We have already previously reported that the transition state of ATP hydrolysis is accessible for DnaB by employing the ATP-analogue ADP:AlF₄⁻⁵⁷, but the direct identification of hydrogen bonds required for characterizing the noncovalent interactions driving molecular recognition of both, ATP and DNA, was hardly possible in a systematic manner and only unspecific spatial proximities derived from ³¹P-¹³C/¹⁵N correlation experiments or the proton chemical-shift values were explored⁴⁴.”

“Note, that we herein report the first proton-detected ³¹P,¹H correlation spectrum at fast MAS frequencies using a sub-milligram sample amount, which was so far not possible with any of the previous equipment. This is an important new step for proving hydrogen bonding in protein-nucleic acid complexes ranging from proteins involved in DNA replication or virus assemblies by solid-state NMR and to derive nucleotide binding modes, even in quite large systems as the one we looked at.”

Significance. The authors showcase a wide array of magnetic resonance methods applied to the DnaB model ATPase. As such, this work is of interest in particular to the solid-state NMR and EPR communities. Taking an inventory of all the heteronuclei that occur in the transition state analogue, the study is a very solid contribution to the field, albeit not radically new.

In terms of structural biology, the manuscript validates published crystal structures of related homologue DnaB proteins and adds a few novel details that are inaccessible to crystallographic studies.

Noteworthy results. As a new feature of the transition state, the authors discover that the AlF₄⁻ ion, which is thought to mimic the trigonal-bipyramidal state of the phosphoryl transfer reaction, undergoes fast reorientation.

Contacts between the protein, specifically the Walker A motif, and ADP as well as the phosphate backbone of the DNA substrate are identified.

The documentation of the methods is sufficient for a specialized readership and the conclusions of the authors are mostly supported by the data.

Major points:

1. The crystal structures of DnaB homologues and other P-loop ATPases form the basis for interpreting the NMR data: assignment of cross peaks, DFT calculations. Given the central importance of these structures for the understanding of this study, it would be important that the authors clearly state in the introduction what structures have been previously published and how

they compare to and possibly differ from the transition state of HpDnaB. This would allow the reader to judge where it is appropriate to draw analogies and mingle structural analysis.

We thank the reviewer for this helpful advice and have inserted a discussion of the two available crystal structures of SF4-type helicases complexed with ATP-analogues (and DNA) already in the introduction and mention the similarity of the reported *Bst*DnaB complex with the complex we have studied. A detailed comparison is then performed in the Discussion Section.

“*Hp*DnaB belongs to the class of SF4-type, ring-shaped DnaB helicases, and only two crystal structures in complex with ATP-analogues have been reported so far, namely the one from *Aquifex aeolicus*²¹ (*Aa*DnaB, PDB accession code 4NMN) *Bacillus stearothermophilus*²² (currently *Geobacillus stearothermophilus*, *Bst*DnaB, PDB 4ESV). *Aa*DnaB is complexed with ADP only and *Bst*DnaB with GDP:AlF₄⁻ as well as single-stranded DNA which is similar to our ADP:AlF₄⁻ complex that we study in presence and absence of DNA herein.”

2. For example, the DFT calculations are based on crystal structures. The authors draw the conclusion that the coordination mode of the Mn²⁺ ion must be similar in HpDnaB. Could the authors identify and test alternative structures in which this ion sits closer to the alpha-phosphate? If there are no examples of such alternative geometries, one might argue that the question answered was not a pressing one in the first place.

The DFT calculations were performed to get an idea if the two ³¹P hyperfine coupling constants observed are in agreement with the two crystal structures available for SF4 helicases that show a coordination of the metal ion to the Pβ of ADP, which is in contrast to ADP:Mn²⁺ complexes in solution for which a symmetric coordination has been described (our current reference 65 Potapov, A. & Goldfarb, D. *Appl. Magn. Reson.* 30, 461 (2006)). A detailed experimental and DFT calculation study has also been published by Un et al. (Un, S. & Bruch, E.M. *Inorg. Chem.* 54, 10422-10428 (2015)) focusing on the influence of Mn²⁺ coordination to phosphates on the ³¹P hyperfine coupling constants. We have included this citation now also in our manuscript.

We agree with the reviewer that the coordination of Mn²⁺ to the Pβ of ADP is probably expected, since the metal ion in ATPases is typically expected to coordinate to the Pβ and Pγ of ATP. We already cite a comparable work by EPR (current reference 93 Kaminker, I., Sushenko, A., Potapov, A., Daube, S., Akabayov, B., Sagi, I. et al. *J. Am. Chem. Soc.* 133, 15514-15523 (2011)). We however decide to leave the DFT calculations in the manuscript, since they support our assignments and the identification of the Mn²⁺ binding mode even if this is only one (minor) aspect of our work.

3. The authors employ three methods to detect hydrogen bonds: chemical shifts, hPH correlation experiments and temperature coefficients of chemical shifts. A table listing each putative hydrogen bond, which methods can detect it, and whether the hydrogen bond is compatible with crystal structure(s) would be handy for the reader.

We thank the reviewer for the suggestion and have included a new Table 1 in the manuscript summarizing the identification of the H-bonds by the different spectroscopic parameters. We have added the following sentence to the manuscript:

“Hydrogen bonds were identified spectroscopically by combining the information of high-frequency shifted proton resonances, spatial proximities probed in hPH correlation experiments and proton chemical shift temperature coefficients (see Table 1 for a summary).”

4. The authors find that the AlF₄ rotates rapidly and is not rigidified by coordinating side chains. On

the other hand, the moiety is immobilized to such an extent that crystallization is feasible and NMR samples are stable for days if not weeks. Could the authors comment on the stability of the complex and on motions other than rotational diffusion of the AIF4 unit?

We have not crystallized our complexes and instead studied sedimented protein samples. We believe that the sedimentation technique is particular useful for studying such systems, especially compared to X-ray crystallography. We have recently thoroughly studied (Wiegand, T., Lacabanne, D., Torosyan, A., Boudet, J., Cadalbert, R., Allain, F.H. et al. *Front Mol Biosci* 7, 17 (2020).) the stability of sedimented protein samples and in the case of the DnaB:ADP:AIF4::DNA complex we for instance have observed that the complex is stable for years in such a sediment, probably associated with the high protein concentration (around 400 mg/mL). We have stated that more clearly in the article now.

“A key advantage of solid-state NMR is the straightforward sample preparation, which simply consists of sedimentation from solution into the solid-state NMR rotor without requiring crystallization step^{53,54} yielding long-term stable protein sample⁵⁵.”

There are further protein motions present in the investigated complex, e.g. the largest fraction of the N-terminal domain undergoes large motions, possibly on the microsecond timescale, since it escapes our cross-polarization NMR spectra as mentioned in Wiegand, T., Cadalbert, R., Lacabanne, D., Timmins, J., Terradot, L., Bockmann, A. et al. *Nat. Commun.* 10, 31 (2019).

5. The ¹⁹F chemical shifts of AIF4⁻ in the absence and presence of DNA in Figure 4 are not identical while the shifts of the free species are well reproducible. Could the authors comment on this difference?

The reviewer is right and we attribute this effect still to small structural differences between the complex in presence and absence of DNA as reflected in protein chemical-shift perturbations described previously (see for example Figure 3b in Wiegand, T., Cadalbert, R., Lacabanne, D., Timmins, J., Terradot, L., Bockmann, A. et al. *Nat. Commun.* 10, 31 (2019)). ¹⁹F NMR seems to be quite sensitive to detect these changes. The ³¹P chemical shifts of ADP in contrast are the same for the two complexes indicating that the structural differences are minor.

6. In Figure 5, the authors speculate that snapshots obtained from multiple crystal structures of different P-loop NTPases retrace a series of reaction steps, from early to late transition state. Could the authors provide concrete evidence or test this hypothesis?

The structure of the AIF4⁻-mimicked early transition state, which we modelled in Figure 5a, although absent for SF4 helicases, is similar in a plethora of diverse P-loop NTPases. Therefore, we envision a similar structure for the DnaB helicase. The structures of the first and second late transition states in Figures 5b and 5c, respectively, are directly taken from the only AIF4⁻-containing DnaB structure, as obtained in the Steitz lab (Itsathitphaisarn, O., Wing, Richard A., Eliason, William K., Wang, J. & Steitz, Thomas A. *Cell* 151, 267-277 (2012). In the revised manuscript we discuss in more details the structures of H₂PO₄⁻ containing post-catalytic states in myosin and emphasize their difference from the AIF4⁻-containing state in HpDnaB (see the additional section included in the Supplementary Information).

For example: In Figure 6, the authors project their model of the nucleotide-binding domain of HpDnaB onto the crystal structure of BstDnaB. Could this model be validated with point mutants of

critical residues, for example the activating lysine and arginine residues, in order to manipulate the reaction? Another possibility would be to align the sequences of the homologues, identify critical amino-acid substitutions and introduce these into HpDnaB. If the hypothesis of the authors holds, one would expect that the mobility of the AlF₄ moiety could be tuned in this manner. As a last resort, the authors could simply perform ¹⁹F and ²⁷Al NMR spectroscopy on trapped transition states of those homologues that they predict feature earlier or later transition states. One would expect to observe a difference in the mobility of AlF₄.

This is indeed a highly interesting idea and we plan to study further protein systems complexed with ADP:AlF₄⁻ in the next future by NMR and EPR. However, this is more a long-term project from both, the biochemical and spectroscopic view point. Particularly, we need to establish for each system under which conditions (and if at all) suitable samples for solid-state NMR are accessible.

REVIEWER COMMENTS

Reviewer #1 (Remarks to the Author):

see attached

Reviewer #3 (Remarks to the Author):

In the revision the authors have extended their discussion and elaborated on the implication of their NMR data in the context of available crystal structures that mimic the transition state of DnaB.

It is a little disappointing that the authors have decided not to pursue any of the suggested additional NMR experiments. More effort would be required to properly validate the author's hypothesis as to where the ADP.AIF4⁻ mimic would be placed with respect to the ATP-hydrolysis coordinate. This is unfortunate because such a mechanistic understanding would have given the manuscript some novelty.

In view of the speculative nature of the current functional analysis of the ATPase enzyme, and the modest advance in NMR methodology with respect to the authors' previous extensive work, my overall recommendation is to publish the manuscript in a more specialized journal, in agreement with reviewer 2.

While the authors have addressed most of my concerns I find their response lacking for some important points as described below. These should be properly addressed before the manuscript can be accepted.

1. Point 1. Please indicate specifically that the Mn(II) substitution did not harm the activity on HpDnaB. As is the statement sounds too general.
2. The new ENDOR spectrum (S1) shows a splitting of about 2 MHz, I agree that this indicates proximity but the authors could go a bit deeper in their analysis. Although the SNR is not great it is clear that the doublet also has a lineshape that could be analyzed and isotropic and anisotropic hyperfine couplings can be evaluated and some structural information can be derived.
3. Regarding point 5 – the authors state **“Since in these experiments the protein has natural isotope abundance, there are no protein-specific resonances (such as ¹⁵N or ¹³C) to be evaluated. As can be expected, there are ¹⁴N resonances visible in EDNMR below 30 MHz that are not shown because this region naturally suffers from strong peak overlap which reduces the attainable information.”** ¹⁴N and ¹⁵N signals have been reported for EDNMR of Mn(II) so it is possible to see them. NMR measurements were carried out on a ¹³C-¹⁵N labeled protein so EDNMR experiments should have been done on this sample and then seeing the ¹³C and ¹⁵N signals give very strong evidence that at least part of the Mn(II) is coordinated to the protein. One of the virtues of this manuscript is the combination of the NMR and EPR methods, yet no advantage has been taken regarding sample availability.
4. Point # 6. The broad feature in the purple spectrum in Fig. 1a masks the ¹⁹F signal and therefore no conclusions can be made that it indeed lacks a ¹⁹F signal. Similarly, there seem to be a weak signal at the ²⁷Al position in this spectrum. So the conclusion that ²⁷Al and ¹⁹F signals cannot come from a free Mn(II)-ADP-^{AlF}₄⁻ is ambiguous and so is the statement **“To corroborate that these resonances are due to DnaB-bound Mn₂₊:ADP:AlF₄⁻ and to rule out that these correlations originate from the formation of the Mn₂₊:ADP:AlF₄⁻ complex in solution, we recorded EDNMR spectra on a frozen control solution in the absence of protein and indeed we do not observe any ¹⁹F and ²⁷Al resonances (purple spectrum in Figure 1a).”** I looked in the ref the authors indicate and copied Fig 4a below. I do not see any such signal in the spectra. There is a very weak broad feature more similar to that observed in the cyan spectrum of the current work, which does not mask the ¹⁹F signal. The same holds for the second ref provided by the authors. I also do not understand what the authors mean by a Mn(II) hyperfine signal – this should be explained. Indeed sometimes the EDNMR spectra show a line at the hyperfine coupling and this was observed for nitroxide. The hyperfine coupling of Mn(II) is around 280 MHz, not 130 MHz.

5. I looked at ref 64 from where the cyan spectrum was taken. Here the doublet around 38 MHz is assigned to ^{13}C whereas in the original paper it is marked as ^{23}Na . This deserves a comment.
6. Point 8 – please state this clearly in the manuscript as the reader maybe wondering.

Reviewer1:

While the authors have addressed most of my concerns I find their response lacking for some important points as described below. These should be properly addressed before the manuscript can be accepted.

1. Point 1. Please indicate specifically that the Mn(II) substitution did not harm the activity on HpDnaB. As is the stament sounds too general.

We believe that the current wording in the manuscript (“The native Mg²⁺ cofactor is replaced for such studies by the EPR-observable paramagnetic Mn²⁺ analogue²⁹ (the biological functionality is maintained to about 80% under such conditions compared to the one observed in presence of Mg²⁺ ³⁰) reflects best the findings described in reference 30. The DNA unwinding activity drops upon magnesium substitution by only 20% as the authors describe in their work and thus does not reach 100% activity and we believe it is important to mention that.

2. The new ENDOR spectrum (S1) shows a splitting of about 2 MHz, I agree that this indicate proximity but the authors could go a bit deeper in their analysis. Although the SNR is not great it is clear that the doublet also has a lineshape that could be analyzed and isotropic and anisotropic hyperfine couplings can be evaluated and some structural information can be derived.

We thank the reviewer for the suggestion and have analysed the line shape of the ¹⁹F ENDOR spectrum in Figure S1. We have included a brief discussion of the resulting parameters and distances (as well as their ambiguities due to the limited signal-to-noise ratio) in the caption of Figure S1.

3. Regarding point 5 – the authors state “**Since in these experiments the protein has natural isotope abundance, there are no protein-specific resonances (such as ¹⁵N or ¹³C) to be evaluated. As can be expected, there are ¹⁴N resonances visible in EDNMR below 30 MHz that are not shown because this region naturally suffers from strong peak overlap which reduces the attainable information.**” ¹⁴N and ¹⁵N signals have been reported for EDNMR of Mn(II) so it is possible to see them. NMR measurements were carried out on a ¹³C-¹⁵N labeled protein so EDNMR experiments should have been done of this sample and then seeing the ¹³C and ¹⁵N signals give very strong evidence that at least part of the Mn(II) is coordinated to the protein. One of the virtues of this manuscript is the combination of the NMR and EPR methods, yet no advantage has been taken regarding sample availability.

We thank the reviewer for the advice and have now measured EDNMR spectra of the DnaB:Mn²⁺:ADP:AlF₄⁻ complex using ¹³C/¹⁵N labeled protein (Figure S3). Additional reference spectra in absence of protein were measured as well. The spectrum in presence of the protein now reveals intense ¹³C and ¹⁵N signals assigned to isotope-labeled DnaB. This further corroborates our conclusion that Mn²⁺:ADP:AlF₄⁻ binds to the protein, particularly also in light of the spectrum obtained in the absence of protein in which the ¹³C/¹⁵N peaks

due to the protein are, as expected, missing. Besides these additional $^{13}\text{C}/^{15}\text{N}$ resonances observed using $^{13}\text{C}/^{15}\text{N}$ labeled protein, the spectrum looks similar, e.g. also the ^{19}F , ^{27}Al and ^{31}P resonances were observed as discussed already in the manuscript.

We have added the spectra to the manuscript now (Figure S3) and discuss them also in the main text.

“We additionally performed EDNMR experiments using uniformly $^{13}\text{C}/^{15}\text{N}$ labeled DnaB complexed with $\text{Mn}^{2+}:\text{ADP}:\text{AlF}_4^-$. While the same ^{19}F , ^{27}Al and ^{31}P features discussed above are present in the spectrum, additional intense ^{13}C and ^{15}N resonances are observed (Figure S3). In combination with the absence of such resonances in the corresponding reference spectrum measured in the absence of protein (Figure S3), this provides further evidence for binding of $\text{Mn}^{2+}:\text{ADP}:\text{AlF}_4^-$ to DnaB.”

4. Point # 6. The broad feature in the purple spectrum in Fig. 1a masks the ^{19}F signal and therefore no conclusions can be made that it indeed lacks a ^{19}F signal. Similarly, there seem to be a weak signal at the ^{27}Al position in this spectrum. So the conclusion that ^{27}Al and ^{19}F signals cannot come from a free $\text{Mn}(\text{II})\text{-ADPAIF}_4^-$ is ambiguous and so is the statement **“To corroborate that these resonances are due to DnaB-bound $\text{Mn}^{2+}:\text{ADP}:\text{AlF}_4^-$ and to rule out that these correlations originate from the formation of the $\text{Mn}^{2+}:\text{ADP}:\text{AlF}_4^-$ complex in solution, we recorded EDNMR spectra on a frozen control solution in the absence of protein and indeed we do not observe any ^{19}F and ^{27}Al resonances (purple spectrum in Figure 1a).”** I looked in the ref the authors indicate and copied Fig 4a below. I do not see any such signal in the spectra. There is a very weak broad feature more similar to that observed in the cyan spectrum of the current work, which does not mask the ^{19}F signal. The same holds for the second ref provided by the authors. I also do not understand what the authors mean by a $\text{Mn}(\text{II})$ hyperfine signal – this should be explained. Indeed sometimes the EDNMR spectra show a line at the hyperfine coupling and this was observed for nitroxide. The hyperfine coupling of $\text{Mn}(\text{II})$ is around 280 MHz, not 130 MHz.

First, regarding the broad Mn^{2+} -related signal marked with * in Figure 1, we added more detail and a reference to the caption of Figure 1, stating, “* marks a broad, currently unassigned, feature that is possibly due to Mn double quantum or combination lines²³”. Similarly, we state in the caption of Figure S3, “The * marks a broad, currently unassigned feature that could either be due to Mn double quantum transitions or combination lines, or simply a baseline artefact due to the high power HTA pulses.”

Given these hypotheses, if a ^{19}F resonance would be present at the same time as the unassigned peak discussed here, we expect the ^{19}F peak (unless severely broadened) to appear as an additional (additive) change in echo intensity, which is not observed here in any of the cases other than $\text{Mn}^{2+}:\text{DnaB}:\text{ADP}:\text{AlF}_4^-$.

That the peak in question is also present in the absence of ^{19}F is further illustrated in the figure here below.

Figure: W-band EDNMR spectra from Figure 1 (cyan and purple, using high power HTA pulses) and additionally the spectrum of Mn²⁺:DnaB:ADP:Vi, a vanadate-trapping control (acquired and shown partly in D. Lacabanne et al., *Molecules* 2020, 25, 5268; doi:10.3390/molecules25225268) is shown. All three EDNMR spectra show the peak marked with * with varying intensities, also in the absence of ¹⁹F. While we do not have a definite assignment based on the present data, we hypothesize the peak could either be due to Mn double quantum transitions or combination lines, or simply a baseline artefact due to the high power HTA pulses.

Additionally, we have specified the HTA pulse power levels termed “high” and “low” power in the Methods section.

5. I looked at ref 64 from where the cyan spectrum was taken. Here the doublet around 38 MHz is assigned to ¹³C whereas in the original paper it is marked as ²³Na. This deserves a comment.

We thank the reviewer for carefully checking the Figure and have included now also ²³Na in Figure 1 as a further assignment possibility for the resonance at around 38 MHz.

6. Point 8 – please state this clearly in the manuscript as the reader maybe wondering.

We have included a statement now in the manuscript.

“Note that EPR experiments were performed on the protein complex in absence of DNA in contrast to most solid-state NMR experiments described below. As described in earlier work, the ADP:AlF₄⁻ states in presence and absence of DNA are highly similar⁵⁷ and we thus recorded EPR experiments only on one of these complexes.”

Reviewer #3 (Remarks to the Author):

In the revision the authors have extended their discussion and elaborated on the implication of their NMR data in the context of available crystal structures that mimic the transition state of DnaB.

It is a little disappointing that the authors have decided not to pursue any of the suggested additional NMR experiments. More effort would be required to properly validate the author's hypothesis as to where the ADP:AlF₄⁻ mimic would be placed with respect to the ATP-hydrolysis coordinate.

This is unfortunate because such a mechanistic understanding would have given the manuscript some novelty.

In view of the speculative nature of the current functional analysis of the ATPase enzyme, and the modest advance in NMR methodology with respect to the authors' previous extensive work, my overall recommendation is to publish the manuscript in a more specialized journal, in agreement with reviewer 2.

In our opinion, the investigation and characterization of further homologues and/or mutations of DnaB complexed with ADP:AlF₄⁻ goes beyond the scope of the current article where we present for the first time how such a transition state trapped for a large motor protein can be studied by EPR and solid-state NMR. We describe novel NMR strategies (e.g. temperature-dependend proton chemical-shift values and the first reported proton-detected ³¹P, ¹H correlation spectrum at fast MAS) that will pave the way for further structural studies of protein-nucleic acid complexes. Our study shows that solid-state NMR spectra at fast magic-angle spinning frequencies possess the resolution to identify hydrogen bonds even in an extremely large system (488 residues/monomer), an information hardly accessible by other structural means than NMR. Additionally, we present for the first time dynamic insights into the rotation of AlF₄⁻ by state-of-the-art heteronuclear MAS experiments showing for this particular case the rotation of the AlF₄⁻ unit, an information which is crucial for the application of such ATP-mimics in structural biology. We agree with the reviewer that solid-state NMR is the key technique that will reveal new insights into the dynamic properties of such ATP-mimics and their location on the ATP hydrolysis reaction coordinate, which requires extensive additional work on further ATPase homologues we will focus on in the next years.

REVIEWER COMMENTS

Reviewer #1 (Remarks to the Author):

The authors have addressed all my concerns, the new spectra added to the SI are important additions to the manuscript and I have no further concerns and recommend publication of the revised version.